# PERSONALIZED SUBGRAPH FEDERATED LEARNING

## ABSTRACT

In real-world scenarios, subgraphs of a larger global graph may be distributed across multiple devices or institutions, and only locally accessible due to privacy restrictions, although there may be links between them. Recently proposed subgraph Federated Learning (FL) methods deal with those missing links across private local subgraphs while distributively training Graph Neural Networks (GNNs) on them. However, they have overlooked the inevitable heterogeneity among subgraphs, caused by subgraphs comprising different communities of a global graph, therefore, consequently collapsing the incompatible knowledge from local GNN models trained on heterogeneous graph distributions. To overcome such a limitation, we introduce a new subgraph FL problem, personalized subgraph FL, which focuses on the joint improvement of the interrelated local GNN models rather than learning a single global GNN model, and propose a novel framework, *FEDerated Personalized sUBgraph learning* (FED-PUB), to tackle it. A crucial challenge in personalized subgraph FL is that the server does not know which subgraph each client has. FED-PUB thus utilizes functional embeddings of the local GNNs using random graphs as inputs to compute similarities between them, and use them to perform weighted averaging for server-side aggregation. Further, it learns a personalized sparse mask at each client to select and update only the subgraph-relevant subset of the aggregated parameters. We validate FED-PUB for its subgraph FL performance on six datasets, considering both non-overlapping and overlapping subgraphs, on which ours largely outperforms relevant baselines.

## 1 INTRODUCTION

Most of the previous Graph Neural Networks (GNNs) (Hamilton, 2020) focus on a single graph, whose nodes and edges collected from multiple sources are stored in a central server. For instance, in a social network platform, every user, with his/her social networks, contributes to creating a giant network consisting of all users and their connections. However, in some practical scenarios, each user/institution collects its own private graph, which is only locally accessible due to privacy restrictions. For instance, as described in Zhang et al. (2021), each hospital may have its own patient interaction network to track their physical contacts or co-diagnosis of a disease, however, such a graph may not be shared with others. How can we then collaboratively train, without sharing actual data, a neural network with its subgraphs distributed across multiple participants (i.e., clients)? The most straightforward way is to perform Federated Learning (FL) with GNNs. Specifically, each client will individually train a local GNN on the private local data, while a central server aggregates locally updated GNN weights from multiple clients into one, and then transmits it back to the clients.

However, an important challenge for such the subgraph FL scenario is how to deal with potentially *missing edges* between subgraphs that are not captured by individual data owners, but may carry important information (See Figure 1 (A)). Recent subgraph FL methods (Wu et al., 2021a; Zhang et al., 2021) additionally tackle this problem by expanding the local subgraph from other subgraphs, as illustrated in Figure 1 (B). In particular, they expand the local subgraph either by exactly augmenting the relevant nodes from the other subgraphs at the other clients (Wu et al., 2021a), or by estimating the nodes using the node information in the other subgraphs (Zhang et al., 2021). However, such sharing of node information may compromise data privacy and can incur high communication costs.

Also, there exists a more important challenge that has been overlooked by existing subgraph FL. We observe that they suffer from large performance degeneration (See Figure 1 right), due to the *heterogeneity* among subgraphs, which is natural since subgraphs comprise different parts of a global

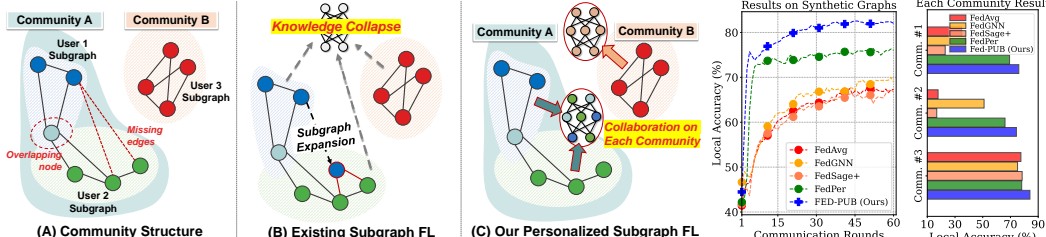

Figure 1: **(A) An illustration of local subgraphs distributed across multiple participants** with overlapping nodes, missing edges and community structures between subgraphs. **(B) Existing subgraph FL methods** (Wu et al., 2021a; Zhang et al., 2021) expand the local subgraphs to tackle the missing edge problem, but collapse incompatible knowledge from heterogeneous subgraphs. **(C) Our personalized subgraph FL** focuses on the joint improvement of local models working on interrelated subgraphs, such as ones within the same community, by selectively sharing the knowledge across them. **(Right:) Knowledge collapse results**, where local models belonging to two small communities (Communities 1 and 2) suffer from large performance degeneration by existing subgraph FL, such as FedGNN (Wu et al., 2021a; 2022) and FedSage+ (Zhang et al., 2021). A personalized FL method, FedPer (Arivazhagan et al., 2019), also underperforms ours since it only focuses on individual model's improvement without sharing local personalization layers between similar subgraphs.

graph. Specifically, two individual subgraphs – for example, User 1 and 3 subgraphs in Communities A and B respectively in Figure 1 (A) – are sometimes completely disjoint having opposite properties. Meanwhile, two densely connected subgraphs form a community (e.g., User 1 and 2 subgraphs within the Community A of Figure 1 (A)), in which they share similar characteristics. However, it is challenging to consider such heterogeneity arising from community structures of a graph.

Motivated by this challenge, we introduce a novel problem of personalized subgraph FL, whose goal is to jointly improve the interrelated local models trained on the interconnected local subgraphs, for instance, subgraphs belonging to the same community, by sharing weights among them (See Figure 1 (C)). However, implementing such selective weight sharing is challenging, since we do not know which subgraph each client has, due to its local accessibility. To resolve this issue, we use functional embeddings of GNNs on random graphs to obtain similarity scores between two local GNNs, and then use them to perform weighted averaging of the model weights at the server. However, the similarity scores only tell how relevant each local model from the other clients is, but not which of the parameters are relevant. Thus we further learn and apply personalized sparse masks on the local GNN at each client to obtain only the subnetwork, relevant to the local subgraph. We refer to this subgraph FL framework as *FEDerated Personalized sUBgraph learning* (FED-PUB).

We extensively validate our FED-PUB on six different datasets with varying numbers of clients, under both overlapping and disjoint subgraph FL scenarios. The experimental results show that ours significantly outperforms relevant baselines. Further analyses show that our method can discover community structures among subgraphs, and the masking scheme localizes the knowledge with respect to the subgraph of each client. Our main contributions are as follows:

- We introduce a novel problem of personalized subgraph FL, which aims at collaborative improvements of the related local models (e.g. subgraphs belonging to the same community), which has been relatively overlooked by previous approaches on graph and subgraph FL.

- We propose a novel framework for personalized subgraph FL, which performs weighted averaging of the local model parameters based on their functional similarities obtained without accessing the data, and learns sparse masks to select only the relevant subnetworks for the given subgraphs.

- We validate our framework on six real-world datasets under both overlapping and non-overlapping node scenarios, demonstrating its effectiveness over existing subgraph FL baselines.

## 2 RELATED WORK

**Graph Neural Networks**  Graph Neural Networks (GNNs) (Hamilton, 2020; Zhou et al., 2020; Wu et al., 2021b; Jo et al., 2021; Baek et al., 2021), which aim to learn the representations of nodes, edges, and entire graphs, are an extensively studied topic. Most existing GNNs under a message passing scheme (Gilmer et al., 2017) iteratively represent a node by aggregating features from its neighboring nodes as well as itself. For example, Graph Convolutional Network (GCN) (Kipf & Welling, 2017) approximates the spectral graph convolutions (Hammond et al., 2011), yielding a mean aggregation over neighboring nodes. Similarly, for each node, GraphSAGE (Hamilton et al., 2017) aggregates the features from its neighbors to update the node representation. While they lead

to successes on node classification and link prediction tasks for single graphs, they are not directly applicable to real-world systems with locally distributed graphs, where graphs from different sources are not shared across participants, which gives rise to federated learning approaches to train GNNs.

**Federated Learning** Federated Learning (FL) (Li et al., 2021) is an essential approach for our distributed subgraph learning problem. To mention a few, FedAvg (McMahan et al., 2017) locally trains a model for each client and then transmits the trained model to a server, while the server aggregates the model weights from local clients and then sends the aggregated model back to them. However, since the locally collected data from different clients may largely vary, heterogeneity is a crucial issue. To tackle this, FedProx (Li et al., 2020) proposes the regularization term that minimizes the weight differences between local and global models, which prevents the model from diverging to the local training data. However, when the local data is extremely heterogeneous, it is more appropriate to collaboratively train a personalized model for each client rather than learning a single global model. FedPer (Arivazhagan et al., 2019) is such a method, which shares only the base layers while having local personalized layers for each client, to keep the local knowledge. Furthermore, to share the learned knowledge between heterogeneous clients, recent studies propose to distill the outputs from clients (Lin et al., 2020; Sattler et al., 2021; Zhu et al., 2021), or directly minimize the differences of their model outputs (Makhija et al., 2022). However, unlike the commonly studied image and text data, graph-structured data is defined by connections between instances, and consequently introduces additional challenges: missing edges and shared nodes between private subgraphs.

**Graph Federated Learning** Few recent studies propose to use the FL framework to collaboratively train GNNs without sharing graph data (He et al., 2021), which can be broadly classified into subgraph- and graph-level methods. Graph-level FL methods assume that different clients have completely disjoint graphs (e.g., molecular graphs), and recent work (Xie et al., 2021; He et al., 2022) focuses on the heterogeneity among non-IID graphs (i.e., difference in graph labels across various clients). In contrast to graph-level FL methods that have similar challenges to general FL scenarios, the subgraph-level FL problem we target has a unique graph-structural challenge, that there exists missing yet probable links between subgraphs, since a subgraph is a part of a larger global graph. To deal with such a missing link problem among subgraphs, existing methods (Wu et al., 2021a; Zhang et al., 2021) augment the nodes by requesting the node information in the other subgraphs, and then connecting the existing nodes with the augmented ones. However, this scheme may compromise data privacy constraints, and also increases communication overhead across clients. Unlike existing subgraph FL that focuses on the problem of missing links, our subgraph FL method tackles the problem with a completely different perspective, focusing on exploring subgraph communities (Girvan & Newman, 2002; Radicchi et al., 2004), which are groups of densely connected subgraphs.

## 3 PROBLEM STATEMENT

We provide general descriptions of Graph Neural Networks (GNNs) and Federated Learning (FL), and then define our novel problem of personalized subgraph FL lying at the intersection of them.

**Graph Neural Networks** A graph $\mathcal{G} = (\mathcal{V}, \mathcal{E})$ consists of a set of $n$ nodes $\mathcal{V}$ and a set of $m$ edges $\mathcal{E}$ along with its node feature matrix $\boldsymbol{X} \in \mathbb{R}^{n \times d}$, where each row represents a $d$-dimensional node feature. $(u, v) \in \mathcal{E}$ represents an edge from a node $u$ to a node $v$. Then, GNNs (Hamilton, 2020) generally represent each node based on features from its neighbors as well as itself, as follows:

$$\boldsymbol{H}_v^{l+1} = \text{UPDATE}^l \left( \boldsymbol{H}_v^l, \text{AGGREGATE}^l \left( \{ \boldsymbol{H}_u^l : \forall u \in \mathcal{N}(v) \} \right) \right), \tag{1}$$

where $\boldsymbol{H}_v^l$ is the feature matrix for node $v$ at $l$-th layer, $\mathcal{N}(v)$ denotes a set of adjacent nodes of node $v$: $\mathcal{N}(v) = \{u \in \mathcal{V} \mid (u, v) \in \mathcal{E}\}$, AGGREGATE aggregates the features of $v$'s neighbors, and UPDATE updates the node $v$'s representation given its previous representation and the aggregated representations from the neighbors. $\boldsymbol{H}^1$ is initialized as input node features $\boldsymbol{X}$.

**Federated Learning** The goal of FL is to collaboratively train a model with local data. Let assume we have $K$ clients with locally collected data inaccessible from others: $\mathcal{D}_k = \{\boldsymbol{X}_i, \boldsymbol{y}_i\}_{i=1}^{N_k}$, where $\boldsymbol{X}_i$ is a data instance, $\boldsymbol{y}_i$ is its corresponding class label, and $N_k$ is the number of data instances at $k$-th client. Then, a popular FL algorithm, FedAvg (McMahan et al., 2017), works as follows:

1. **(Initialization)** At the initial communication round $r = 0$, the central server first selects $K$ clients that are available for training, and initializes their local model parameters as the global parameter $\bar{\boldsymbol{\theta}}$, represented as follows: $\boldsymbol{\theta}_k^{(0)} \leftarrow \bar{\boldsymbol{\theta}}^{(0)} \ \forall k$, where $\boldsymbol{\theta}_k^{(0)}$ is the parameters for $k$-th client.

2. **(Local Updates)** Each active local model performs training on private local data $\mathcal{D}_k$ to minimize the task loss $\mathcal{L}(\mathcal{D}_k; \boldsymbol{\theta}_k^{(0)})$, consequently updating the parameters $\boldsymbol{\theta}_k^{(1)} \leftarrow \boldsymbol{\theta}_k^{(0)} - \eta \nabla \mathcal{L}$.

3. **(Global Aggregation)** After local training, the server aggregates the locally learned knowledge with respect to the number of training instances, i.e., $\bar{\boldsymbol{\theta}}^{(1)} \leftarrow \frac{N_k}{N} \sum_{k=1}^{K} \boldsymbol{\theta}_k^{(1)}$ with $N = \sum_k N_k$, and distributes the updated global parameters $\bar{\boldsymbol{\theta}}^{(1)}$ to the local clients selected at the next round.

This FL algorithm iterates between Step 2 and 3 until reaching the final round $R$.

**Challenges in Subgraph FL** While the above FL works well on image and text data, due to the unique structure of graphs, there exist nontrivial challenges for applying this FL scheme to graph-structured data. In particular, unlike with an image domain where each instance $\boldsymbol{X}_i$ is independent to the other images, each node $v$ in a graph is always influenced by its relationships to adjacent nodes $\mathcal{N}(v)$. Moreover, a local graph $G_i$ could be a subgraph of a larger global graph $\mathcal{G}$: $G_i \subseteq \mathcal{G}$. In such a case, there could be missing edges between subgraphs in two different clients: $(u, v)$ with $u \in \mathcal{V}_i$ and $v \in \mathcal{V}_j$ for clients $i$ and $j$, respectively. To tackle this problem, existing methods (Wu et al., 2021a; Zhang et al., 2021) estimate the nodes of a local subgraph $G_k$ based on the node information from subgraphs at other clients $G_i \,\forall i \neq k$, and then extend the existing nodes with the estimated ones. However, this augmentation scheme incurs high communication costs as it requires sharing node information across clients, which may also violate data privacy constraints (Abadi et al., 2016).

Yet, there exists a more challenging issue. Assume that we have a global graph consisting of all subgraphs. Then, there are *communities* of such subgraphs (Radicchi et al., 2004; Girvan & Newman, 2002; Porter et al., 2009), where subgraphs within the same community are more densely connected to each other than subgraphs outside the community. Formally, a global graph $\mathcal{G}$ can be decomposed into $T$ different communities: $C_i \subseteq \mathcal{G} \,\forall i = 1, ..., T$, where $i$-th community $C_i = (\mathcal{V}_i, \mathcal{E}_i)$ consists of densely connected nodes. Then, in a subgraph FL problem, a local subgraph $G_j$ belongs to at least one community: $C_i = \bigcup_{j=1}^{J} G_j$. Note that, based on the theory of network homophily (McPherson et al., 2001), such connected subgraphs within the same community have similar properties, while subgraphs in two opposite communities are not. Such distributional heterogeneity of communities may lead a naive FL algorithm to collapse incompatible knowledge across different communities.

**Personalized Subgraph FL** To prevent the above knowledge collapse issue, we aim to personalize the subgraph FL algorithm by performing personalized weight averaging of local model parameters; thereby capturing the community structure among interrelated subgraphs. To be formal, the objective of existing subgraph FL (Wu et al., 2021a; Zhang et al., 2021; Liu et al., 2021) is as follows: $\min_{\bar{\boldsymbol{\theta}}} \sum_{G_i \subseteq \mathcal{G}} \mathcal{L}(G_i; \bar{\boldsymbol{\theta}})$. However, finding a universal set of parameters (i.e., $\bar{\boldsymbol{\theta}}$) that works on all tasks will result in finding a suboptimal parameter set, since subgraphs in two different communities with sparse connections are extremely heterogeneous due to the network homophily. To address this limitation, we formulate a novel problem of personalized subgraph FL, formalized as follows:

$$\min_{\{\boldsymbol{\theta}_i, \boldsymbol{\mu}_i\}_{i=1}^{K}} \sum_{G_i \subseteq \mathcal{G}} \mathcal{L}(G_i; \boldsymbol{\theta}_i, \boldsymbol{\mu}_i), \; \boldsymbol{\theta}_i \leftarrow \boldsymbol{\mu}_i \odot \left( \sum_{j=1}^{K} \alpha_{ij} \boldsymbol{\theta}_j \right) \text{ with } \alpha_{ik} \gg \alpha_{il} \text{ for } G_k \subseteq C \text{ and } G_l \nsubseteq C, \quad (2)$$

where $\boldsymbol{\theta}_i$ is the weight for subgraph $G_i$ belonging to community $C$. $\alpha_{ij}$ is the coefficient for weight aggregation between clients $i$ and $j$, which can promote the collaborative learning across multiple local models working on interrelated subgraphs that belong to the same community, by assigning larger weights on them. However, this scalar coefficient $\alpha_{ij}$ cannot inform us which elements of the aggregated weight are relevant to subgraph $G_i$. Therefore, we further multiply it to the trainable sparse vector $\boldsymbol{\mu}_i$ with element-wise multiplication $\odot$, to shift and filter out irrelevant weights from subgraphs of heterogeneous communities. We will specify how to obtain $\alpha$ and $\boldsymbol{\mu}$ in Section 4.

## 4 FEDERATED PERSONALIZED SUBGRAPH LEARNING FRAMEWORK

To realize our goal of personalized subgraph FL (equation 2), we propose to compute subgraph similarities for detecting communities, and to mask weights from subgraphs in unrelated communities.

### 4.1 SUBGRAPH SIMILARITY ESTIMATION FOR DETECTING SUBGRAPH COMMUNITY

We aim to capture the community structure consisting of a group of densely connected subgraphs. Note that, due to network homophily where similar instances in the graph are more associated with

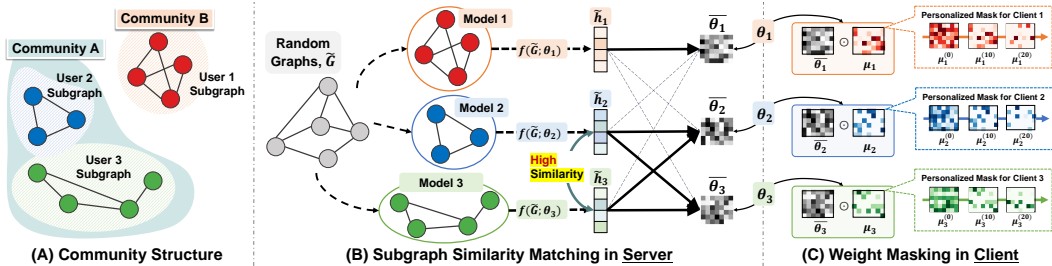

Figure 2: **(A) Two communities**, where Community A and B consist of one and two subgraphs, respectively. **(B) Client Similarity Matching**: we first forward randomly generated graphs to models $f(\tilde{G}; \theta_i)$, and obtain functional embeddings $\tilde{h}_i$, which are then used to estimate subgraph similarities. Then, the similarities are used in weight aggregation, resulting in personalized model weights $\bar{\theta}_i$. **(C) Weight Masking**: transmitted weights from the server to clients $\bar{\theta}_i$ are masked and shifted by local masks $\mu_i$ for localization to the local subgraph.

each other (McPherson et al., 2001), subgraphs within the same community should be similar. In other words, if one can measure subgraph similarities, we can group similar ones into the community. However, measuring similarity between local subgraphs is challenging since we do not know which subgraph each client has due to local accessibility. How can we then compute subgraph similarities, without accessing them? To this end, we propose to approximate the similarity at local clients using auxiliary information obtained from the local GNN models working on the subgraphs.

**Subgraph Similarity Estimation with Model Parameters**   For measuring the similarity between local subgraphs, without accessing them, we may use the model parameters as proxies, as follows: $S(i,j) = (\theta_i \cdot \theta_j)/(\|\theta_i\|\|\theta_j\|)$, where $\theta$ is a parameter flatten into a vector, and $S$ is a similarity measure. This may sound reasonable since the GNN model trained on the subgraph will embed its knowledge into its parameters. However, this scheme has a notable drawback that similarity measured in the high-dimensional parameter space is not meaningful due to the curse of dimensionality (Bellman, 1966), and that the cost of calculating the similarity between parameters grows rapidly as the model size increases (See Figure 3).

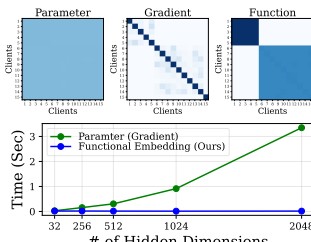

Figure 3: Effectiveness (top) and efficiencies (bottom) of different similarity measurements.

**Subgraph Similarity Estimation with Functional Embedding**   To tackle the limitation of using parameter distance, we propose to measure the functional similarity of neural networks by feeding the same input to every local model and then calculating the similarities using their outputs, inspired by neural network search (Jeong et al., 2021). The main intuition is that we can consider the transformation defined with a neural network as a function, and we measure the functional similarity of two networks by the distance of their outputs for the same input. However, unlike the previous work, which uses Gaussian noises as inputs for image classification, we use random graphs as inputs as we work with GNNs. Formally, let $\tilde{G} = (\tilde{\mathcal{V}}, \tilde{\mathcal{E}})$ be a random community graph designed by a stochastic block model (Holland et al., 1983), where subgraphs within the community have more edges between them than edges across the communities (See Appendix B.3 for initialization details). Then, the similarity between two functions defined by GNNs at clients $i$ and $j$ is defined as follows:

$$S(i,j) = \frac{\tilde{h}_i \cdot \tilde{h}_j}{\|\tilde{h}_i\|\|\tilde{h}_j\|}, \quad \tilde{h}_i = \text{AVG}(f(\tilde{G}; \theta_i)) \text{ and } \tilde{h}_j = \text{AVG}(f(\tilde{G}; \theta_j)), \tag{3}$$

where $\tilde{h}$ is the averaged output of all node embeddings for input $\tilde{G}$ with average operation, AVG. We provide additional discussions with results on similarity estimation in Appendix C.6 and C.7.

**Personalized Weight Aggregation with Subgraph Similarity**   With equation 3, the remaining step is then to share the model weights between models working on similar subgraphs belonging to the same community. However, entirely ignoring model parameters from different communities may result in exploiting only the local objective while ignoring globally useful weights, which results in suboptimal performance (See Appendix C.8 for details). Therefore, we perform weighted averaging of local models from all clients based on their functional similarities, as follows (Figure 2 (B)):

$$\bar{\theta}_i \leftarrow \sum_{j=1}^{K} \alpha_{ij} \cdot \theta_j, \quad \alpha_{ij} = \frac{\exp(\tau \cdot S(i,j))}{\sum_k \exp(\tau \cdot S(i,k))}, \tag{4}$$

where $\alpha_{ij}$ is a normalized similarity between clients $i$ and $j$, and $\tau$ is a hyperparameter for scaling the unnormalized similarity score. Note that increasing the value of $\tau$ (e.g., 10) will result in model averaging done almost exclusively among subgraphs detected as belonging to the same community.

This personalized scheme handles two challenges in subgraph FL. First, in contrast to global weight aggregation which collapses the knowledge from heterogeneous communities, our subgraph FL allows the models belonging to different communities to obtain individual parameters that are beneficial for each community. Also, missing edges (i.e., a lack of information sharing) between interconnected subgraphs, which are explicitly handled by expanding local subgraphs in existing works (Wu et al., 2021a; Zhang et al., 2021), could be implicitly considered by largely sharing the knowledge among models of probably linked subgraphs within the same community (See Figure 6 and 9). This enhances data privacy while minimizing communication costs between subgraphs.

## 4.2 ADAPTIVE WEIGHT MASKING FOR SELECTING SUBGRAPH-RELEVANT PARAMETERS

Based on the previous similarity matching scheme, we can effectively group GNNs that belong to the same community, thus preventing the collapsing of irrelevant knowledge from other communities. However, the scalar weighting scheme only considers how much each local model from other clients is relevant for the subgraph task, but not *which* parameters are relevant. Thus we propose a scheme to select only the relevant parameters from the aggregated model weights transmitted from the server.

**Personalized Parameter Masking**  We perform selective training and updating of the aggregated parameters by modulating and masking them, using sparse local masks (Figure 2 (C)). Formally, let $\boldsymbol{\mu}_i$ be a local mask for client $i$. Then, our local model weight is obtained by modulating the weights from the server, as follows: $\boldsymbol{\theta}_i = \bar{\boldsymbol{\theta}}_i \odot \boldsymbol{\mu}_i$, where $\odot$ is an element-wise multiplication operation between the globally given weight $\bar{\boldsymbol{\theta}}_i$ and the local mask $\boldsymbol{\mu}_i$. Note that $\boldsymbol{\mu}_i$ is a free variable and not shared across clients. Also, we initialize $\boldsymbol{\mu}_i$ as ones, in order to start training with the globally initialized model parameters without modification. We then further promote sparsity on the mask, to take two advantages. First, we can transmit only the partial parameters, that have not been sparsified at the client, to the server rather than sending all parameters, thus reducing the communication costs. Also, if local masks are sufficiently sparse, local models can be trained faster, when zero-skipping operations are supported. To take these benefits in sparsity, we use $L_1$ regularizer on $\boldsymbol{\mu}_i$ when performing local optimization (See Appendix B.3 for details on sparsification), shown in equation 5.

**Preventing Local Divergence with Proximal Term**  As masks are trained only with limited local data without parameter sharing, they may be easily overfitted to the training instances in each client. To alleviate this issue, we adopt the proximal term proposed in Li et al. (2020) that regularizes the locally updated model $\boldsymbol{\theta}_i$ to be closer to the globally given model $\bar{\boldsymbol{\theta}}_i$, therefore, preventing the model from extremely drifting to the local training distribution. To sum up, at $i$-th client, our objective function including sparsity and proximal terms with $L_1$ and $L_2$ losses is denoted as follows:

$$\min_{(\boldsymbol{\theta}_i, \boldsymbol{\mu}_i)} \mathcal{L}(G_i; \boldsymbol{\theta}_i, \boldsymbol{\mu}_i) + \lambda_1 \|\boldsymbol{\mu}_i\|_1 + \lambda_2 \|\boldsymbol{\theta}_i - \bar{\boldsymbol{\theta}}_i\|_2^2, \tag{5}$$

where $\mathcal{L}$ is a conventional cross-entropy loss function, and $\lambda_1$ and $\lambda_2$ are scaling hyper-parameters.

## 5 EXPERIMENTS

We now experimentally validate our FED-PUB on six different datasets under both the overlapping and disjoint subgraph scenarios with varying client numbers, on node classification tasks.

### 5.1 EXPERIMENTAL SETUPS

**Datasets**  Following the experimental setup from Zhang et al. (2021), we construct distributed subgraphs from the benchmark dataset by dividing it into the number of participants: each FL participant has a subgraph that is a part of an original graph. In particular, we use six datasets: Cora, CiteSeer, Pubmed and ogbn-arxiv for citation graphs (Sen et al., 2008; Hu et al., 2020); Computer and Photo for product graphs (McAuley et al., 2015; Shchur et al., 2018). We then divide the original graph into multiple subgraphs using the METIS graph partitioning algorithm (Karypis, 1997). Note that, unlike the Louvain algorithm (Blondel et al., 2008), used in Zhang et al. (2021), that requires to further merge partitioned subgraphs into particular numbers of subgraphs since it cannot specify the number of subsets (i.e., clients for FL), the METIS algorithm can specify the number of subsets,

Table 1: **Results on the overlapping node scenario.** The reported results are mean and standard deviation over three different runs. The statistically significant performances ($p > 0.05$) are emphasized in bold.

| | Cora | | | CiteSeer | | | Pubmed | | | - |
|---|---|---|---|---|---|---|---|---|---|---|
| Methods | 10 Clients | 30 Clients | 50 Clients | 10 Clients | 30 Clients | 50 Clients | 10 Clients | 30 Clients | 50 Clients | - |
| Local | 73.98 ± 0.25 | 71.65 ± 0.12 | 76.63 ± 0.10 | 65.12 ± 0.08 | 64.54 ± 0.42 | 66.68 ± 0.44 | 82.32 ± 0.07 | 80.72 ± 0.16 | 80.54 ± 0.11 | - |
| FedAvg | 76.48 ± 0.36 | 53.99 ± 0.98 | 53.99 ± 4.53 | 69.48 ± 0.15 | 66.15 ± 0.64 | 66.51 ± 1.00 | 82.67 ± 0.11 | 82.05 ± 0.12 | 80.24 ± 0.35 | - |
| FedProx | 77.85 ± 0.50 | 51.38 ± 1.74 | 56.27 ± 9.04 | 69.39 ± 0.35 | 66.11 ± 0.75 | 66.53 ± 0.43 | 82.63 ± 0.17 | 82.13 ± 0.13 | 80.50 ± 0.46 | - |
| FedPer | 78.73 ± 0.31 | 74.18 ± 0.24 | 74.42 ± 0.37 | 69.81 ± 0.28 | 65.19 ± 0.81 | 67.64 ± 0.44 | 85.31 ± 0.06 | 84.35 ± 0.38 | 83.94 ± 0.10 | - |
| GCFL | 78.84 ± 0.26 | 73.41 ± 0.27 | 76.63 ± 0.16 | 69.48 ± 0.39 | 64.92 ± 0.18 | 65.98 ± 0.30 | 83.59 ± 0.25 | 80.77 ± 0.12 | 81.36 ± 0.11 | - |
| FedGNN | 70.63 ± 0.83 | 61.38 ± 2.33 | 56.91 ± 0.82 | 68.72 ± 0.39 | 59.98 ± 1.52 | 58.98 ± 0.98 | 84.25 ± 0.07 | 82.02 ± 0.22 | 81.85 ± 0.10 | - |
| FedSage+ | 77.52 ± 0.46 | 51.99 ± 0.42 | 55.48 ± 11.5 | 68.75 ± 0.48 | 65.97 ± 0.02 | 65.93 ± 0.30 | 82.77 ± 0.08 | 82.14 ± 0.11 | 80.31 ± 0.68 | - |
| FED-PUB (Ours) | **79.60 ± 0.12** | **75.40 ± 0.54** | **77.84 ± 0.23** | **70.58 ± 0.20** | **68.33 ± 0.45** | **69.21 ± 0.30** | **85.70 ± 0.08** | **85.16 ± 0.10** | **84.84 ± 0.12** | - |

| | Amazon-Computer | | | Amazon-Photo | | | ogbn-arxiv | | | All |
|---|---|---|---|---|---|---|---|---|---|---|
| Methods | 10 Clients | 30 Clients | 50 Clients | 10 Clients | 30 Clients | 50 Clients | 10 Clients | 30 Clients | 50 Clients | Avg. |
| Local | 88.50 ± 0.20 | 86.66 ± 0.00 | 87.04 ± 0.02 | 92.17 ± 0.12 | 90.16 ± 0.12 | 90.42 ± 0.15 | 62.52 ± 0.07 | 61.32 ± 0.04 | 60.04 ± 0.04 | 76.72 |
| FedAvg | 88.99 ± 0.19 | 83.37 ± 0.47 | 76.34 ± 0.12 | 92.91 ± 0.07 | 89.30 ± 0.22 | 74.19 ± 0.57 | 63.56 ± 0.02 | 59.72 ± 0.06 | 60.94 ± 0.24 | 73.38 |
| FedProx | 88.84 ± 0.20 | 83.84 ± 0.89 | 76.60 ± 0.47 | 92.67 ± 0.19 | 89.17 ± 0.40 | 72.36 ± 2.06 | 63.52 ± 0.11 | 59.86 ± 0.16 | 61.12 ± 0.04 | 73.38 |
| FedPer | 89.30 ± 0.04 | 87.99 ± 0.23 | 88.22 ± 0.27 | 92.88 ± 0.24 | 91.23 ± 0.16 | 90.92 ± 0.38 | 63.97 ± 0.08 | 62.29 ± 0.04 | 61.24 ± 0.11 | 78.42 |
| GCFL | 89.01 ± 0.22 | 87.24 ± 0.09 | 87.02 ± 0.22 | 92.45 ± 0.10 | 90.58 ± 0.11 | 90.54 ± 0.08 | 63.24 ± 0.02 | 61.66 ± 0.10 | 60.32 ± 0.01 | 77.61 |
| FedGNN | 88.15 ± 0.09 | 87.00 ± 0.10 | 83.96 ± 0.88 | 91.47 ± 0.11 | 87.91 ± 1.34 | 78.90 ± 6.46 | 63.08 ± 0.19 | 60.09 ± 0.04 | 60.51 ± 0.11 | 73.66 |
| FedSage+ | 89.24 ± 0.15 | 81.33 ± 1.20 | 76.72 ± 0.39 | 92.76 ± 0.05 | 88.69 ± 0.99 | 72.41 ± 1.36 | 63.24 ± 0.02 | 59.90 ± 0.12 | 60.95 ± 0.09 | 73.12 |
| FED-PUB (Ours) | **89.98 ± 0.08** | **89.15 ± 0.06** | **88.76 ± 0.14** | **93.22 ± 0.07** | **92.01 ± 0.07** | **91.71 ± 0.11** | **64.18 ± 0.04** | **63.34 ± 0.12** | **62.55 ± 0.12** | 79.53 |

(a) Cora    (b) CiteSeer    (c) Pubmed    (d) Computer    (e) Photo    (f) ogbn-arxiv

Figure 4: **Convergence plots for the overlapping node scenario.** We visualize the test accuracy curves for all six datasets corresponding to Table 1, over 100 communication rounds with 30 clients.

thus making more reasonable experimental settings in subgraph FL (See Appendix C.2). For the non-overlapping scenario where there are no duplicate nodes between subgraphs, we use the output from the METIS as it provides non-overlapping partitions. Meanwhile, for the overlapping scenario where nodes are duplicated among subgraphs, we randomly sample the subsets (i.e., subgraphs) of the partitioned graph multiple times. For more details on datasets, please refer to Appendix B.1.

**Baselines and Our Model** **1) FedAvg** (McMahan et al., 2017) and **2) FedProx** (Li et al., 2020): The most popular FL baselines. **3) FedPer** (Arivazhagan et al., 2019): A personalized FL baseline without sharing personalized layers. **4) FedGNN (FedPerGNN)**[1] (Wu et al., 2021a; 2022) and **5) FedSage+** (Zhang et al., 2021): Subgraph FL baselines which we mainly target. **6) GCFL** (Xie et al., 2021): A graph FL baseline which works on completely disjoint graphs (i.e., graph-level FL) as in clustered FL (Sattler et al., 2020), adopted for subgraph FL. **7) Local**: A baseline without sharing weights with other clients. **8) FED-PUB**: Our personalized subgraph FL including subgraph similarity matching and weight masking. We provide further descriptions in Appendix B.2.

**Implementation Details** We use the two layer GCNs (Kipf & Welling, 2017) as the base GNN for all models. We perform FL over 100 communication rounds for Cora, CiteSeer and Pubmed datasets, while 200 rounds for Computer, Photo and arxiv datasets, considering the size of datasets. The local training epoch is selected in the range of $\{1, 2, 3\}$ depending on the dataset size (e.g., Computer is three while CiteSeer is one)[2]. We use the Adam optimizer Kingma & Ba (2015) for model optimization. We then measure the node classification accuracy on subgraphs at the client-side, and then average the performance across clients. See Appendix B.3 for more details.

## 5.2 EXPERIMENTAL RESULTS

**Main Results** Table 1 shows the node classification performance under the overlapping subgraph scenario, in which our FED-PUB statistically ($p > 0.05$) significantly outperforms all the baselines. In particular, while FedGNN and FedSage+ are two pioneer works for the subgraph FL problem, they significantly underperform personalized FL methods including ours, especially at the larger number of clients. This is even surprising as they share node information between clients for handling the missing edge problem, yet we suppose such inferior performance comes from naive averaging of local weights without consideration of community structures. While personalized FL baselines including FedPer and GCFL show decent performance by alleviating the knowledge collapse issue between subgraphs with local parameterization or clustering, they still largely underperform ours as they are not concerned with aggregation between similar subgraphs that form a community (i.e., GCFL uses a bi-partitioning scheme, which iteratively divides a group of subgraphs within the same

---

[1]FedGNN is extended to FedPerGNN, where the core algorithm of averaging all client gradients is the same.
[2]We found communication rounds and local epochs are important factors to prevent overfitting of all models.

Table 2: **Results on the non-overlapping node scenario.** The reported results are mean and standard deviation over three different runs. The statistically significant performances ($p > 0.05$) are emphasized in bold.

| Methods | Cora | | | CiteSeer | | | Pubmed | | | - |
| | 5 Clients | 10 Clients | 20 Clients | 5 Clients | 10 Clients | 20 Clients | 5 Clients | 10 Clients | 20 Clients | - |
|---|---|---|---|---|---|---|---|---|---|---|
| Local | 81.30 ± 0.21 | 79.94 ± 0.24 | 80.30 ± 0.25 | 69.02 ± 0.05 | 67.82 ± 0.13 | 65.98 ± 0.17 | 84.04 ± 0.18 | 82.81 ± 0.39 | 82.65 ± 0.03 | - |
| FedAvg | 74.45 ± 5.64 | 69.19 ± 0.67 | 69.50 ± 3.58 | 71.06 ± 0.60 | 63.61 ± 3.59 | 64.68 ± 1.83 | 79.40 ± 0.11 | 82.71 ± 0.29 | 80.97 ± 0.26 | - |
| FedProx | 72.03 ± 4.56 | 60.18 ± 7.04 | 48.22 ± 6.81 | 71.73 ± 1.11 | 63.33 ± 3.25 | 64.85 ± 1.35 | 79.45 ± 0.25 | 82.55 ± 0.24 | 80.50 ± 0.25 | - |
| FedPer | 81.68 ± 0.40 | 79.35 ± 0.04 | 78.01 ± 0.32 | 70.41 ± 0.32 | 70.53 ± 0.28 | 66.64 ± 0.27 | 85.80 ± 0.21 | 84.20 ± 0.28 | 84.72 ± 0.31 | - |
| GCFL | 81.47 ± 0.65 | 78.66 ± 0.27 | 79.21 ± 0.70 | 70.34 ± 0.57 | 69.01 ± 0.12 | 66.33 ± 0.05 | 85.14 ± 0.33 | 84.18 ± 0.19 | 83.94 ± 0.36 | - |
| FedGNN | 81.51 ± 0.68 | 70.12 ± 0.99 | 70.10 ± 3.52 | 69.06 ± 0.92 | 55.52 ± 3.17 | 52.23 ± 6.00 | 79.52 ± 0.23 | 83.25 ± 0.45 | 81.61 ± 0.59 | - |
| FedSage+ | 72.97 ± 5.94 | 69.05 ± 1.59 | 57.97 ± 12.6 | 70.74 ± 0.69 | 65.63 ± 3.10 | 65.46 ± 0.74 | 79.57 ± 0.24 | 82.62 ± 0.31 | 80.82 ± 0.25 | - |
| FED-PUB (Ours) | **83.70 ± 0.19** | **81.54 ± 0.12** | **81.75 ± 0.56** | **72.68 ± 0.44** | **72.35 ± 0.53** | **67.62 ± 0.12** | **86.79 ± 0.09** | **86.28 ± 0.18** | **85.53 ± 0.30** | - |

| Methods | Amazon-Computer | | | Amazon-Photo | | | ogbn-arxiv | | | All |
| | 5 Clients | 10 Clients | 20 Clients | 5 Clients | 10 Clients | 20 Clients | 5 Clients | 10 Clients | 20 Clients | Avg. |
|---|---|---|---|---|---|---|---|---|---|---|
| Local | 89.22 ± 0.13 | 88.91 ± 0.17 | 89.52 ± 0.20 | 91.67 ± 0.09 | 91.80 ± 0.02 | 90.47 ± 0.15 | 66.76 ± 0.07 | 64.92 ± 0.09 | 65.06 ± 0.05 | 79.57 |
| FedAvg | 84.88 ± 1.96 | 79.54 ± 0.24 | 74.79 ± 0.24 | 89.89 ± 0.83 | 83.15 ± 3.71 | 81.35 ± 1.04 | 65.54 ± 0.07 | 64.44 ± 0.10 | 63.24 ± 0.13 | 74.58 |
| FedProx | 85.25 ± 1.27 | 83.81 ± 1.09 | 73.05 ± 1.30 | 90.38 ± 0.48 | 80.92 ± 4.64 | 82.32 ± 0.29 | 65.21 ± 0.20 | 64.37 ± 0.18 | 63.03 ± 0.04 | 72.84 |
| FedPer | 89.67 ± 0.34 | 89.73 ± 0.04 | 87.86 ± 0.43 | 91.44 ± 0.37 | 91.76 ± 0.23 | 90.59 ± 0.06 | 66.87 ± 0.05 | 64.99 ± 0.18 | 64.66 ± 0.11 | 79.94 |
| GCFL | 89.07 ± 0.91 | 90.03 ± 0.16 | 89.08 ± 0.25 | 91.99 ± 0.29 | 92.06 ± 0.25 | 90.79 ± 0.17 | 66.80 ± 0.12 | 65.09 ± 0.08 | 65.08 ± 0.04 | 79.90 |
| FedGNN | 88.08 ± 0.15 | 88.18 ± 0.41 | 83.16 ± 0.13 | 90.25 ± 0.70 | 87.12 ± 2.01 | 81.00 ± 4.48 | 65.47 ± 0.22 | 64.21 ± 0.32 | 63.80 ± 0.05 | 75.23 |
| FedSage+ | 85.04 ± 0.61 | 80.50 ± 1.30 | 70.42 ± 0.85 | 90.77 ± 0.44 | 76.81 ± 8.24 | 80.58 ± 1.15 | 65.69 ± 0.09 | 64.52 ± 0.14 | 63.31 ± 0.20 | 73.47 |
| FED-PUB (Ours) | **90.74 ± 0.05** | **90.55 ± 0.13** | **90.12 ± 0.09** | **93.29 ± 0.19** | **92.73 ± 0.18** | **91.92 ± 0.12** | **67.77 ± 0.09** | **66.58 ± 0.08** | **66.64 ± 0.12** | **81.59** |

| (a) Cora | (b) CiteSeer | (c) Pubmed | (d) Computer | (e) Photo | (f) ogbn-arxiv |

Figure 5: **Convergence plots for the non-overlapping node scenario.** We visualize the test accuracy curves for all six datasets corresponding to Table 2, over 100 communication rounds with 10 clients.

community into two disjoint sets). We then further conduct the experiments on the disjoint subgraph scenario (i.e., non-overlapping scenario), where nodes are not overlapped between subgraphs, which makes the subgraph FL problem more heterogeneous. As shown in Table 2, FED-PUB consistently outperforms all existing baselines in such a challenging scenario, demonstrating the efficacy of ours.

**Fast Local Convergence** As shown in Figure 4 and 5, our FED-PUB converges rapidly, compared against baselines including personalized FL models. We conjecture that this is because, not only ours can accurately identify subgraphs forming the community and then share weights substantially across them for promoting the joint improvement, but also masking out subgraph-irrelevant weights received from the server for localization to local subgraphs, demonstrated in the next two paragraphs.

**Community Detection** We aim to show whether the proposed FED-PUB can group subgraphs comprising a community during the personalized weight aggregation. Note that, if two different subgraphs have many missing edges or have similar label distributions, we usually consider those two as within the same community (Radicchi et al., 2004; Girvan & Newman, 2002; Porter et al., 2009). Thereby, as shown in Figure 6 (a) and (b), there are four different communities by the interval of five, and the last two communities further comprise a larger community. Then, as shown in Figure 6 (c) and (d), our FED-PUB detects obvious four communities at the first few rounds, and then captures the larger yet somewhat less-obvious community consisting of two smaller communities.

**Ablation Study** To analyze the contribution of each component, we conduct ablation studies. As shown in Figure 7, we observe that each of our subgraph similarity matching and weight masking schemes significantly improves the performances from the naive FedAvg, while the performance is much improved when using both together. However, the benefit from each component is different across overlapping and non-overlapping scenarios. In particular, in the former scenario where a group of densely overlapped subgraphs usually comprise a community, similarity matching for community detection is more beneficial since capturing the community would promote the joint improvement of subgraphs belonging to the same community. However, in the non-overlapping scenario, two individual subgraphs become more heterogeneous, thus selectively using the aggregated model weights from the server with personalized weight masks improves the performance a lot (See additional results and discussions on heterogeneity with sparse weight masks in Appendix C.4).

**Communication Efficiency** Another notable advantage of using sparse masks is that we can reduce the communication costs at every FL round, as well as the model size for faster training. In particular, as demonstrated in Table 3, existing subgraph FL methods require more than two times larger communications costs, measured by adding both the client-to-server and server-to-client costs, compared against the naive FedAvg, since they require to transfer additional node information between clients for estimating the probable nodes on the subgraphs. Contrarily, our FED-PUB has significantly lower communication costs and lower model sizes by using sparse masks on model

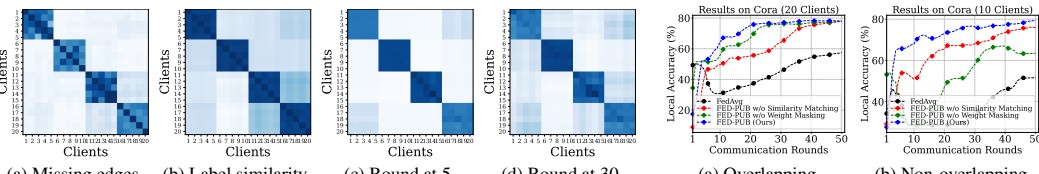

| (a) Missing edges | (b) Label similarity | (c) Round at 5 | (d) Round at 30 |

Figure 6: **The heatmaps of the community structure** on overlapping node scenario with Cora (20 clients). Dark color indicates lots of missing edges between subgraphs (a) or high similarities in labels (b). (c) and (d) are functional similarities captured by our FED-PUB.

| (a) Overlapping | (b) Non-overlapping |

Figure 7: **Ablation studies** of the proposed FED-PUB on both overlapping (a) and non-overlapping (b) subgraph scenarios, on the Cora dataset.

| Model | Acc. [%] | Model Size [%] | Cost [%] |
|---|---|---|---|
| FedAvg | $76.48 \pm 0.36$ | $100.00 \pm 0.00$ | $100.00 \pm 0.00$ |
| FedGNN | $70.63 \pm 0.83$ | $100.00 \pm 0.00$ | $214.94 \pm 0.00$ |
| FedSage+ | $77.52 \pm 0.46$ | $100.00 \pm 0.00$ | $276.84 \pm 0.00$ |
| GCFL | $78.84 \pm 0.26$ | $100.00 \pm 0.00$ | $100.00 \pm 0.00$ |
| Ours ($\lambda_1$=9e-1) | $77.36 \pm 0.99$ | $\mathbf{25.13 \pm 0.34}$ | $\mathbf{37.70 \pm 0.56}$ |
| Ours ($\lambda_1$=7e-1) | $79.46 \pm 0.41$ | $42.59 \pm 1.33$ | $63.89 \pm 1.99$ |
| Ours ($\lambda_1$=5e-1) | $\mathbf{79.89 \pm 0.12}$ | $57.07 \pm 0.52$ | $85.61 \pm 0.78$ |

Table 3: **Analysis on efficiencies** of communication costs and model sizes with sparse masks.

Figure 8: **Varying the local epochs** with accuracy curves.

Figure 9: **Performance on interrelated subgraphs**.

weights: transmitting and training only the partial parameters not sparsified at the client. Further, we can manage the trade-off between the model sparsity and the performance by controlling the hyperparameter for sparsity regularization, $\lambda_1$ (See Appendix C.1 for more hyperparameter analyses).

**Varying Local Epochs**   As shown in Figure 8, when we increase the number of communication rounds and the local steps, the model diverges to the local subgraph (i.e., overfitting), due to the small number of training instances and the direct connection between training and test nodes: struggling to generalize to the test instances. However, our model with the proximal term in equation 5 alleviates this issue, therefore, maintaining the highest local performance. Notably, the performance with five local epochs is inferior to the performance of one epoch, which indicates that increasing the local epochs does not always bring advantages and properly tuning them is important for subgraph FL.

**Handling Missing Edges**   The missing edge problem, where two interconnected subgraphs cannot share information due to missing edges between them, is a unique challenge in subgraph FL (See Appendix C.9 for more discussions). To tackle this, existing subgraph FL explicitly augments nodes and edges for capturing the information flow over missing edges between interconnected subgraphs, while ours implicitly shares weights a lot across similar subgraphs within the same community. To measure their efficacy, we evaluate the performance on the neighboring subgraph, which has the most missing edges to the local subgraph, based on its local model weight. Specifically, in Figure 9, (Neighbor) denotes the subgraph performance evaluated by its neighbor model, while (Local) denotes the subgraph performance from its own local model. Then, the high performance on (Neighbor) measure means two associated subgraphs share meaningful knowledge without having actual edges between them, thereby solving the missing edge problem. Figure 9 shows that ours achieves the superior performance on the neighboring subgraph problem against subgraph FL baselines, verifying that ours has an advantage on the missing edge problem by sharing meaningful knowledge between subgraphs having potentially missing edges, without explicitly estimating them.

## 6   CONCLUSION

We introduced a novel problem of personalized subgraph FL, which focuses on the joint improvement of local GNNs working on interrelated subgraphs (e.g. subgraphs belonging to the same community), by selectively utilizing knowledge from other models. The proposed personalized subgraph FL is highly challenging due to 1) difficulty of computing similarities between local subgraphs that are only locally accessible, and 2) knowledge collapse among local models that work on heterogeneous subgraphs during weight aggregation. To this end, we proposed a novel personalized subgraph FL framework, called FEDerated Personalized sUBgraph learning (FED-PUB), which computes the similarities across subgraphs using functional embeddings of their local GNNs on random graphs, and uses them to perform a weighted average of the local models for each client. Further, we mask out globally given weights to focus on only the relevant subnetwork for each community and client. We extensively validated our framework on multiple benchmark datasets with both overlapping and non-overlapping subgraphs, on which our FED-PUB significantly outperforms relevant baselines. Further analyses show the effectiveness of the subgraph similarity matching for detecting the community structures, as well as the weight masking for tackling the subgraph heterogeneity.

REPRODUCIBILITY STATEMENT

We attach the source code of our FED-PUB framework in the supplementary file. Also, we provide every detail of experimental setups including datasets, models, and implementations in Appendix B.

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

Table 4: **Dataset statistics.** We report the number of nodes, edges, classes, clustering coefficient, and heterogeneity for the original graph and its splitted subgraphs on both overlapping and non-overlapping node scenarios. Note that Ori denotes the original graph, and Cli denotes the number of clients.

*Overlapping node scenario*

| | Cora | | | | CiteSeer | | | | Pubmed | | | |
|---|---|---|---|---|---|---|---|---|---|---|---|---|
| | Ori | 10 Cli | 30 Cli | 50 Cli | Ori | 10 Cli | 30 Cli | 50 Cli | Ori | 10 Cli | 30 Cli | 50 Cli |
| # Classes | 7 | | | | 6 | | | | 3 | | | |
| # Nodes | 2,485 | 621 | 207 | 124 | 2,120 | 530 | 177 | 106 | 19,717 | 4,929 | 1,643 | 986 |
| # Edges | 10,138 | 1,249 | 379 | 215 | 7,358 | 889 | 293 | 170 | 88,648 | 10,675 | 3,374 | 1,903 |
| Clustering Coefficient | 0.238 | 0.133 | 0.129 | 0.125 | 0.170 | 0.088 | 0.087 | 0.096 | 0.060 | 0.035 | 0.034 | 0.035 |
| Heterogeneity | N/A | 0.297 | 0.567 | 0.613 | N/A | 0.278 | 0.494 | 0.547 | N/A | 0.210 | 0.383 | 0.394 |

| | ogbn-arxiv | | | | Amazon-Computer | | | | Amazon-Photo | | | |
|---|---|---|---|---|---|---|---|---|---|---|---|---|
| | Ori | 10 Cli | 30 Cli | 50 Cli | Ori | 10 Cli | 30 Cli | 50 Cli | Ori | 10 Cli | 30 Cli | 50 Cli |
| # Classes | 40 | | | | 10 | | | | 8 | | | |
| # Nodes | 169,343 | 42,336 | 14,112 | 8,467 | 13,381 | 3,345 | 1,115 | 669 | 7,487 | 1,872 | 624 | 374 |
| # Edges | 2,315,598 | 282,083 | 83,770 | 44,712 | 491,556 | 59,236 | 16,684 | 8,969 | 238,086 | 29,223 | 8,735 | 4,840 |
| Clustering Coefficient | 0.226 | 0.177 | 0.185 | 0.191 | 0.351 | 0.337 | 0.348 | 0.359 | 0.410 | 0.380 | 0.391 | 0.410 |
| Heterogeneity | N/A | 0.315 | 0.606 | 0.615 | N/A | 0.327 | 0.577 | 0.614 | N/A | 0.306 | 0.696 | 0.684 |

*Non-overlapping node scenario*

| | Cora | | | | CiteSeer | | | | Pubmed | | | |
|---|---|---|---|---|---|---|---|---|---|---|---|---|
| | Ori | 5 Cli | 10 Cli | 20 Cli | Ori | 5 Cli | 10 Cli | 20 Cli | Ori | 5 Cli | 10 Cli | 20 Cli |
| # Classes | 7 | | | | 6 | | | | 3 | | | |
| # Nodes | 2,485 | 497 | 249 | 124 | 2,120 | 424 | 212 | 106 | 19,717 | 3,943 | 1,972 | 986 |
| # Edges | 10,138 | 1,866 | 891 | 422 | 7,358 | 1,410 | 675 | 326 | 88,648 | 16,374 | 7,671 | 3,607 |
| Clustering Coefficient | 0.238 | 0.250 | 0.259 | 0.263 | 0.170 | 0.175 | 0.178 | 0.180 | 0.060 | 0.063 | 0.066 | 0.067 |
| Heterogeneity | N/A | 0.590 | 0.606 | 0.665 | N/A | 0.517 | 0.541 | 0.568 | N/A | 0.362 | 0.392 | 0.424 |

| | ogbn-arxiv | | | | Amazon-Computer | | | | Amazon-Photo | | | |
|---|---|---|---|---|---|---|---|---|---|---|---|---|
| | Ori | 5 Cli | 10 Cli | 20 Cli | Ori | 5 Cli | 10 Cli | 20 Cli | Ori | 5 Cli | 10 Cli | 20 Cli |
| # Classes | 40 | | | | 10 | | | | 8 | | | |
| # Nodes | 169,343 | 33,869 | 16,934 | 8,467 | 13,381 | 2,676 | 1,338 | 669 | 7,487 | 1,497 | 749 | 374 |
| # Edges | 2,315,598 | 410,948 | 182,226 | 86,755 | 491,556 | 84,480 | 36,136 | 15,632 | 238,086 | 43,138 | 19,322 | 8,547 |
| Clustering Coefficient | 0.226 | 0.247 | 0.259 | 0.269 | 0.351 | 0.385 | 0.398 | 0.418 | 0.410 | 0.437 | 0.457 | 0.477 |
| Heterogeneity | N/A | 0.593 | 0.615 | 0.637 | N/A | 0.604 | 0.612 | 0.647 | N/A | 0.684 | 0.681 | 0.751 |

# A ALGORITHMS

In this section, we provide algorithms of the proposed subgraph similarity estimation and adaptive weight masking in our FED-PUB framework. In particular, weight masking, performed in the client, is shown in Algorithm 1. Also, similarity matching, working on the server, is shown in Algorithm 2.

---

**Algorithm 1 FED-PUB** Client Algorithm

1: $R$: the number of rounds, $E$: the number of epochs, $K$: the number of clients, $G_i$: local data for client $i$, $f_i$: model function for client $i$, $\boldsymbol{\theta}_i$: model parameters for client $i$, $\boldsymbol{\mu}_i$: weight masking parameters for client $i$, $S(\cdot)$: similarity matching function, $\tau$: scaling factor for similarity matching.

2: **Function** RunClient($\bar{\boldsymbol{\theta}}_i$)
3: $\boldsymbol{\theta}_i \leftarrow \bar{\boldsymbol{\theta}}_i \odot \boldsymbol{\mu}_i$
4: **for** each local epoch $e$ from 1 to $E$ **do**
5: $\quad \boldsymbol{\theta}_i \leftarrow \boldsymbol{\theta}_i - \eta \nabla \mathcal{L}(G_i; \boldsymbol{\theta}_i, \boldsymbol{\mu}_i)$
6: **end for**
7: **return** $\boldsymbol{\theta}_i$

**Algorithm 2 FED-PUB** Server Algorithm

1: **Function** RunServer()
2: initialize $\bar{\boldsymbol{\theta}}^{(1)}$
3: **for** each round $r = 1, 2, \ldots, R$ **do**
4: $\quad$ **for** $\forall i$ **in parallel do**
5: $\quad\quad$ **if** $r = 1$ **then**
6: $\quad\quad\quad \boldsymbol{\theta}_i^{(r+1)} \leftarrow$ RunClient($\bar{\boldsymbol{\theta}}^{(r)}$)
7: $\quad\quad$ **else**
8: $\quad\quad\quad \bar{\boldsymbol{\theta}}_i^{(r)} \leftarrow \sum_{j=0}^{K} \frac{\exp(\tau \cdot S(i,j))}{\sum_{k=0}^{K} \exp(\tau \cdot S(i,k))} \boldsymbol{\theta}_j$
9: $\quad\quad\quad \boldsymbol{\theta}_i^{(r+1)} \leftarrow$ RunClient($\bar{\boldsymbol{\theta}}_i^{(r)}$)
10: $\quad\quad$ **end if**
11: $\quad$ **end for**
12: **end for**

---

# B EXPERIMENTAL SETUPS

In this section, we first provide the descriptions of six different benchmark datasets that we use, along with their preprocessing setups and statistics for FL in Subsection B.1. Then, we explain the baselines and our proposed FED-PUB in detail in Subsection B.2. Lastly, we further describe the implementation details of experiments on synthetic and real-world graphs, as well as additional experimental details on functional similarities and sparse masks in Subsection B.3.

## B.1 DATASETS

We report statistics of six different benchmark datasets (Sen et al., 2008; Hu et al., 2020; McAuley et al., 2015; Shchur et al., 2018), such as Cora, CiteSeer, Pubmed, and ogbn-arxiv for citation graphs;

Computer and Photo for amazon product graphs, which we use in our experiments for both the overlapping and non-overlapping node scenarios, in Table 4. Specifically, in Table 4, we report the number of nodes, edges, classes, and clustering coefficient for each subgraph, but also the heterogeneity between the subgraphs. Note that, to measure the clustering coefficient, which indicates how much nodes are clustered together, for the individual subgraph, we first calculate the clustering coefficient (Watts & Strogatz, 1998) for all nodes, and then average them. On the other hand, to measure the heterogeneity, which indicates how disjointed subgraphs are dissimilar, we calculate the median Jenson-Shannon divergence of label distributions between all pairs of subgraphs.

For dataset splits, we randomly sample 20% nodes for training, 35% for validation, and 35% for testing, for all datasets except for the arxiv dataset. This is because the arxiv dataset has the relatively larger number of nodes as shown in Table 4, thus we randomly sample 5% nodes for training, the remaining half of the nodes for validation, and the other nodes for testing.

We then describe how to partition the original graph into multiple subgraphs, whose number is the same as the number of clients (i.e., FL participants). In general, we use the METIS graph partitioning algorithm (Karypis, 1997) to divide the original graph into multiple subgraphs, which can control the number of disjoint subgraphs as parameters. Consequently, in the non-overlapping node scenario, the disjoint subgraph for each client is directly obtained by the output of the METIS algorithm (i.e., if we set the parameter value for METIS as 10, then we can obtain 10 different disjoint subgraphs, each of which is given to each client). On the other hand, in the overlapping node scenario where nodes are duplicated across different subgraphs, we first divide the original graph into 2, 6, and 10 disjoint subgraphs for 10 clients, 30 clients, and 50 clients, respectively, with the METIS algorithm. After that, in the one splitted subgraph, we randomly sample half of the nodes and their associated edges, and then use them as the subgraph for one particular client. This procedure is performed five times to generate five different yet overlapped subgraphs, per one splitted subgraph from METIS.

### B.2 BASELINES AND OUR MODEL

1. **FedAvg**: This method (McMahan et al., 2017) is the FL baseline, where each client locally updates a model and sends it to a server, while the server aggregates locally updated models with respect to their numbers of training samples and transmits the aggregated one back to the clients.

2. **FedProx**: This method (Li et al., 2020) is the FL baseline, which prevents the local model from drifting to the local data by minimizing weight differences between local and global models.

3. **FedPer**: This method (Arivazhagan et al., 2019) is the personalized FL baseline, which shares only the base layers, while keeping the personalized classification layers in the local side.

4. **FedGNN**: This method (Wu et al., 2021a) is the subgraph FL baseline, which expands the local subgraph by exactly augmenting the relevant nodes from other clients. In the original paper, the authors consider the nodes, which have shared neighboring nodes, over two individual clients as the relevant nodes, and then augment them. However, in our non-overlapping node scenario, since nodes are unique across different clients, we measure the similarities between nodes in different clients, and then augment them having the similarity above the threshold (e.g., 0.5).

5. **FedSage+**: This method (Zhang et al., 2021) is the subgraph FL baseline, which expands the local subgraph by generating additional nodes with the local graph generator. To train the graph generator, it first transmits the local node representations to other clients, and then calculates the gradient of distances between the transmitted node representations and the other client's node features. Then, the gradient is sent back to the local client, used to train the graph generator.

6. **GCFL**: This method (Xie et al., 2021) is the graph FL baseline, which targets completely disjoint graphs (e.g., molecular graphs) as in image tasks. In particular, it uses the bi-partitioning scheme, which divides a set of clients into two disjoint client groups based on their gradient similarities. Then, the model weights are only shared between grouped clients having similar gradients, after partitioning. Note that this bi-partitioning scheme is similar to the scheme proposed in clustered-FL (Sattler et al., 2020) for image classification, and we adopt this for our subgraph FL.

7. **Local**: This method is the non-FL baseline, which only locally trains the model for each client, and does not share any weights between clients.

8. **FED-PUB**: This is our FEDerated Personalized sUBgraph learning (FED-PUB) framework, which not only estimates the similarity between client subgraphs with their models' functional embeddings for detecting subgraph community structures, but also adaptively masks received weights from the server to filter subgraph-irrelevant weights from heterogeneous communities.

### B.3   IMPLEMENTATION DETAILS

**Implementation Details on Functional Embeddings**   The functional embeddings are key ingredients in the proposed FED-PUB framework, to capture community structures of interconnected subgraphs leveraged in personalized weight aggregation (See Section 4.1). To obtain such the functional embeddings, the graph input of GNNs is required, which we randomly generate via a stochastic block model (Holland et al., 1983). Specifically, we first sample five individual subgraphs, each of which has 100 nodes, in which the probability of edges within the single graph is 0.1, while the probability of edges between different graphs is 0.01. Also, we initialize the node features with the normal distribution of 1.0 mean and 1.0 variance. Note that this randomly sampled graph is initialized at the server-side at once, and the server distributes it to all clients. Then, the client calculates its model's functional embedding, and then transmits only the output embedding to the server.

**Implementation Details on Sparse Masks**   As described in Section 4.2, we propose to sparsify the local personalized mask $\boldsymbol{\mu}_i$ for each client $i$, for taking the benefit in communication and prediction costs. In this paragraph, we additionally provide the detailed implementation specifications on sparse masks during training and test phases of our FED-PUB. First, in training, we regularize the local mask $\boldsymbol{\mu}_k$ to be sparse by minimizing the $L_1$ Norm of it along with its scaling parameter $\lambda_2$ to the local loss $\mathcal{L}$, represented in equation 5. However, this regularization scheme might not be enough to exactly make a subset of local masks zero. Therefore, in the test phase, we use the threshold scheme, where elements (neurons) of $\boldsymbol{\mu}_k$ below a certain threshold (i.e., $\lambda_2$) are set to zero. By doing so, we can transmit only the partial parameters to the server, but also can predict with only the partial parameters, therefore, effectively reducing both communication and prediction costs.

**Common Implementation Details for Experiments**   For all experiments, we stack two layers of Graph Convolutional Network (GCN) (Kipf & Welling, 2017) and one linear classifier layer. Regarding hyperparameters, the number of hidden dimensions is set to 128, and the learning rate is set to 0.001. All models are optimized with Adam optimizer (Kingma & Ba, 2015). Also, all clients participate in FL at every round. For all experiments about our FED-PUB, we set $\lambda_1$ and $\lambda_2$ values for $L_1$ and $L_2$ losses in equation 5 for sparsity and proximal terms as 0.001. While we can tune such two scaling hyperparameters, we observe that those default values show satisfactory performances across all datasets without specific tuning to each dataset (See Appendix C.1 for more analyses).

**Implementation Details on Synthetic Graph Experiments**   We perform two experiments on synthetic graphs, which are shown in Figure 1 and Figure 3. In particular, in the experiment of Figure 1, there are three communities that have different label distributions (e.g., nodes in the first community have label 0, whereas nodes in the last community have label 2), and three communities consist of 5/5/40 non-overlapped subgraphs, with 50 clients. Each subgraph consists of 30 nodes, and the edges between two nodes are sampled from the probability of 0.5. Also, in the experiment of Figure 3, there are two communities that have different label distributions, and two communities have 5/15 non-overlapped subgraphs, with 20 clients. Each subgraph consists of 30 nodes, and the edges between two subgraphs within the same community are sampled from the probability of 0.7, whereas the edges between two subgraphs from different communities are sampled from the probability of 0.01. For all experiments, the number of local epochs is set to 3, and the number of total FL rounds is set to 100. In our FED-PER including its variants of using parameter and gradient for subgraph similarity estimation, the scaling hyperparameter (i.e., $\tau$) for the similarity in equation 4 is set to 10.

**Implementation Details on Real-World Graph Experiments**   For relatively small datasets, namely Cora, CiteSeer and PubMed, we set the number of local training epoch as 1, and the number of total rounds as 100. For larger datasets, such as Computer, Photo and arxiv, we set the number of total rounds as 200, while the number of local epochs is set to 2 for Photo and arxiv, and set to 3 for Computer. In the overlapping node scenario, we set the similarity scaling hyperparameter (i.e., $\tau$) as 5 for all our models. Meanwhile, we set the similarity scaling hyperparameter (i.e., $\tau$) as 3 in the non-overlapping node scenario for all our models. We observe that, the larger $\tau$ value performs better for the overlapping node scenario, in which different subgraphs are easily grouped together, compared to the disjoint node scenario. Finally, we report the test performance of all models at the best validation epoch, and the performance is measured by the node classification accuracy.

**Computing Resources**   For all experiments, we use PyTorch (Paszke et al., 2019) and PyTorch Geometric (Fey & Lenssen, 2019) as deep learning libraries. We use two types of GPUs: GeForce

| $\lambda_1$ | $\lambda_2$ | Accuracy [%] | Sparsity [%] | $\lambda_1$ | $\lambda_2$ | Accuracy [%] | Sparsity [%] |
|---|---|---|---|---|---|---|---|
| 3e-1 | 1e-3 | $79.62 \pm 0.23$ | $28.93 \pm 0.52$ | 7e-1 | 1e-3 | $78.68 \pm 0.59$ | $56.94 \pm 0.29$ |
| 5e-1 | 1e-3 | $79.42 \pm 0.37$ | $42.38 \pm 0.35$ | 7e-1 | 1e-2 | $78.56 \pm 0.05$ | $56.61 \pm 0.32$ |
| 7e-1 | 1e-3 | $78.68 \pm 0.59$ | $56.94 \pm 0.29$ | 7e-1 | 1e-1 | $79.46 \pm 0.41$ | $57.41 \pm 1.33$ |
| 9e-1 | 1e-3 | $77.36 \pm 0.99$ | $74.87 \pm 0.34$ | 7e-1 | 1e-0 | $79.31 \pm 0.45$ | $57.28 \pm 0.16$ |

Figure 10: **Analysis on hyperparameters $\lambda_1$ and $\lambda_2$**, with corresponding model sparsity and performance.

RTX 2080 Ti and TITAN XP, for training models. Note that the runtime of our framework also depends on the number of workers for processing clients' jobs in parallel. In general, we use 10 or 20 workers (i.e., simultaneously training 10 or 20 local models for 10 or 20 clients), and the single run of our algorithm for 50 clients with 1 local epoch and 100 total rounds takes less than 2 hours.

## C   ADDITIONAL EXPERIMENTAL RESULTS

In this section, we provide additional experimental results on sensitive analysis of hyperparameters in Section C.1; varying graph partitioning schemes in Section C.2 and C.3; varying random graph inputs in Section C.6; and varying similarity estimation schemes in Section C.7. In addition to them, we also analyze the subgraph heterogeneity itself in Section C.4 and its relationship to the graph size in Section C.5, as well as the impact of missing edges to the task performance in Section C.9.

### C.1   RESULTS ON VARYING THE SCALING HYPERPARAMETERS IN LOSS FUNCTION

In Figure 10, we explore the effects of hyperparameters $\lambda_1$ and $\lambda_2$ on the Cora dataset with the overlapping node scenario, where the number of local epochs is set as 2 and the number of clients is set as 10. In particular, $\lambda_1$ value can control the degree of the model sparsity, thus, to see its efficacy, we fix $\lambda_2$ value while varying $\lambda_1$, and then measure both the model sparsity and performance. As shown in Figure 10 left, higher $\lambda_1$ values significantly increase the model sparsity, meanwhile, the model performance is slightly decreased. The results indicate that we should consider the trade-off between the sparsity and the model performance, when selecting $\lambda_1$ value. On the other hand, $\lambda_2$ value is designed to prevent the excessive knowledge drift to the local subgraph distribution, and, to verify its effectiveness, we fix $\lambda_1$ value while varying $\lambda_2$. As shown in Figure 10 right, small lambda values lead to the performance degeneration, thus choosing the sufficiently large $\lambda_2$ values (e.g., 1e-1) would yield the high performance. Further, we observe that the sparsity does not depend on $\lambda_2$ value, thus the effects of $\lambda_1$ and $\lambda_2$ are orthogonal and complementary.

### C.2   RESULTS ON LOUVAIN GRAPH PARTITIONING ALGORITHM

To validate our FED-PUB on different graph partitioning settings, we use another experimental setup from Zhang et al. (2021), which uses Louvain algorithm (Blondel et al., 2008) for partitioning the entire graph into several subgraphs for FL clients. Before explaining experimental results, we would like to point out that there is a drawback on the Louvain algorithm presented in Zhang et al. (2021), unlike the METIS algorithm (Karypis, 1997) that we use, for subgraph FL scenarios. Specifically, since the Louvain algorithm cannot spec-

Table 5: Results on experimental settings of Louvain graph partitioning algorithms, following Zhang et al. (2021).

| Methods | Cora | CiteSeer | PubMed |
|---|---|---|---|
| Local | $78.56 \pm 0.27$ | $64.06 \pm 0.09$ | $84.07 \pm 0.17$ |
| FedAvg | $71.83 \pm 0.40$ | $69.23 \pm 0.71$ | $82.47 \pm 0.32$ |
| FedProx | $72.09 \pm 0.29$ | $67.66 \pm 0.97$ | $82.68 \pm 0.34$ |
| FedPer | $80.13 \pm 0.50$ | $66.28 \pm 1.22$ | $85.02 \pm 0.23$ |
| FedGNN | $76.59 \pm 0.66$ | $61.21 \pm 1.46$ | $82.67 \pm 0.26$ |
| FedSage+ | $72.20 \pm 0.60$ | $68.40 \pm 0.61$ | $82.76 \pm 0.09$ |
| GCFL | $78.55 \pm 0.38$ | $64.20 \pm 0.31$ | $84.62 \pm 0.31$ |
| FED-PUB (Ours) | $\mathbf{82.68 \pm 0.13}$ | $\mathbf{69.45 \pm 0.75}$ | $\mathbf{86.20 \pm 0.11}$ |

ify the number of graph partitions, the number of subgraphs on the CiteSeer dataset is 38, where three of them have less than ten nodes. Then, based on those 38 disjoint subgraphs, to generate the particular number of clients (e.g., 10), Zhang et al. (2021) randomly merge the different subgraphs without considering their graph properties. Therefore, even though each partitioned subgraph has its unique structural role/characteristic, the reconstructed 10 subgraphs from the original 38 subgraphs have mixed properties (i.e., two incompatible subgraphs could be merged), which is suboptimal. However, as described in the Datasets paragraph of Section 5.1, the METIS that we use can specify the number of partitions, thus more appropriate for making experimental settings for subgraph FL.

As shown in Table 5, we conduct experiments with the Louvain graph partitioning algorithm (Blondel et al., 2008; Zhang et al., 2021), on Cora, CiteSeer, and PubMed datasets with the number of clients as 10. The results show that our FED-PUB consistently outperforms all the other baselines on the different graph partitioning setting, thus the effectiveness of our FED-PUB becomes obvious.

## C.3    Results on Random Graph Partitioning Algorithm

One might be curious about experiments on uniform partitions of graphs, instead of splitting the graph with sophisticated partitioning algorithms (e.g., METIS and Louvain algorithms). Therefore, in this subsection, we explain why this random partitioning setting is unrealistic, and then show the performances on this random setting. To be specific, if we partition the entire graph of the CiteSeer dataset into different subgraphs uniformly at random, the number of nodes of each subgraph becomes larger than the number of edges (e.g., 211 nodes yet 72 edges per subgraph, thus some nodes do not have any edges),

Table 6: Results on experimental settings of the random graph partitioning.

| Methods | CiteSeer with 10 Clients |
|---|---|
| Local | 44.27 ± 1.05 |
| FedAvg | 60.84 ± 0.80 |
| FedProx | 59.38 ± 1.66 |
| FedPer | 60.04 ± 0.93 |
| FedGNN | 54.64 ± 1.67 |
| FedSage+ | 61.03 ± 0.11 |
| GCFL | 53.15 ± 1.82 |
| FED-PUB (Ours) | **63.63** ± 0.86 |

which is uncommon in practice. However, we further perform experiments on the random split setting with 10 different clients on the CiteSeer dataset, and then report the results in Table 6. As shown in Table 6, the gap between baselines and our model is reduced compared to the non-overlapping and overlapping scenarios in Table 1 and Table 2. This is because there is no specific community structure in this random setting; however, our FED-PUB still consistently outperforms all baselines.

## C.4    Analyses on Distribution Shifts Between Subgraphs with Sparse Masks

To see the distributional shifts between subgraphs in our subgraph FL, we measure label differences between subgraphs with the Jenson-Shannon divergence on the Cora dataset with 20 different clients over the overlapping and non-overlapping scenarios. Then, the results show that the distance (i.e., divergence value) among subgraphs within the same community is 0.384 while the distance between subgraphs belonging to different communities is 0.639 for the non-overlapping node scenario. On the other hand, for the overlapping node scenario, the distance among subgraphs within the same community is 0.047 while the distance between subgraphs belonging to different communities is 0.528. Thus, these results confirm that heterogeneity of subgraphs even within the same community is extremely larger in the non-overlapping setup (0.384) compared to the overlapping setup (0.047).

Then, from the above result, we can further argue that personalized weight aggregation from similarity matching is not enough in disjoint subgraph FL problems, since the model weight received from completely heterogeneous subgraphs might not be meaningful to the local subgraph task, especially in the non-overlapping setting. However, in such the extremely heterogeneous case, a personalized weight masking scheme is obviously helpful, since it can filter out irrelevant information transmitted from the other heterogeneous subgraphs, while allowing the model to maintain the locally helpful information in its parameters. This result is also aligned with the results in Figure 7 of the ablation study that, the personalized weight masking scheme brings huge performance improvements in the non-overlapping setting with high heterogeneity between subgraphs, whereas the personalized weight aggregation scheme is more beneficial in the overlapping setting with low heterogeneity.

Lastly, to directly see the efficacy of sparse masks in subgraph FL, we empirically analyze whether they can indeed filter irrelevant weights received from heterogeneous communities and subgraphs. To do so, we measure how many parameters are shared between the two most dissimilar (i.e., heterogeneous) subgraphs, as well as between the two most similar subgraphs, for the Cora dataset with 20 clients on the non-overlapping node setting. For the two most similar subgraphs within the same community, 75% parameters are shared. Meanwhile, for the two heterogeneous subgraphs from two opposite communities, 73% parameters are filtered by sparse masks. These results demonstrate that sparse masks can prevent the knowledge collapse from subgraphs of heterogeneous communities.

## C.5    Analyses on Local Graph Size vs Heterogeneity

To see how much heterogeneity issues are severe in terms of the number of clients, we first analyze the exact amount of heterogeneities with respect to the client numbers. In particular, following the reported statistics in Table 4, when we increase the number of clients in both the overlapping and non-overlapping node scenarios, the heterogeneity across subgraphs becomes severe and problematic for personalized subgraph FL, and thus becomes an important issue to tackle.

Table 7: Results on Cora, CiteSeer, and PubMed datasets on non-overlapping scenarios, with the number of clients of 3.

| Methods | Cora | CiteSeer | PubMed |
|---|---|---|---|
| Local | 81.73 ± 0.44 | 68.16 ± 0.25 | 84.81 ± 0.40 |
| FedAvg | 78.77 ± 0.13 | 69.34 ± 0.23 | 85.29 ± 0.20 |
| FedProx | 78.91 ± 0.21 | 69.54 ± 0.27 | 85.59 ± 0.18 |
| FedPer | 82.29 ± 0.13 | 69.80 ± 0.33 | 85.34 ± 0.16 |
| FedGNN | 82.36 ± 0.62 | 67.79 ± 0.49 | 85.57 ± 0.13 |
| FedSage+ | 77.79 ± 1.96 | 69.35 ± 0.12 | 85.63 ± 0.22 |
| GCFL | 82.67 ± 0.74 | 68.85 ± 0.58 | 86.20 ± 0.15 |
| FED-PUB (Ours) | **84.45** ± 0.23 | **70.66** ± 0.34 | **86.74** ± 0.16 |

Note that one might be curious about whether our FED-PUB is still effective, when the heterogeneity issue is less significant. Thus, we further conduct the experiment in the setting where the number of clients is 3 on Cora, CiteSeer, and PubMed datasets of the non-overlapping node scenario. As shown in Table 7, compared to the results in Table 2 with client numbers of 5, 10, and 20, the performance gaps between our FED-PUB and baselines are much reduced. However, we can clearly observe that our FED-PUB consistently outperforms all baselines with large margins even when the number of clients is small, since there still exists incompatible knowledge across clients, which our FED-PUB effectively handles with personalized weight aggregation and local weight masking schemes.

## C.6    RESULTS ON VARYING THE GRAPH INPUTS FOR FUNCTIONAL EMBEDDINGS

As described in Section B.3, to obtain the functional embedding, we use the same random graph for all client models, which is initialized by a stochastic block model (Holland et al., 1983) with node features initialized by the normal distribution. The underlying assumption on using the random graph is that such randomness may not yield any bias on the functional space, unlike existing node features of the particular subgraph. In other words, we expect our random graphs are helpful for effectively capturing the similarities among subgraphs.

Table 8: Results on varying the graph inputs for functional embeddings, over overlapping and non-overlapping node scenarios with 20 clients on Cora.

| Graphs | Overlapping | Non-Overlapping |
|---|---|---|
| SBM | 0.937 | 0.810 |
| ER | 0.920 | 0.712 |
| One | 0.822 | 0.656 |
| Feature | 0.897 | 0.632 |

In this subsection, to experimentally validate the above statement, we compare various graph inputs used for calculating the functional embeddings, as follows: 1) SBM denotes the random graph generated by the Stochastic Block Model (SBM) like ours; 2) ER denotes the random graph generated by the Erdos-Renyi (ER) model (Erdős & Rényi, 1960); 3) One denotes the random graph having only one node; 4) Feature denotes the graph where node features are initialized by the existing ones in the client. We then measure the performances of those four schemes by calculating the correlation coefficient between label distributions and estimated similarities of subgraphs (i.e., the high correlation coefficient means that the estimated similarities from functional embeddings are similar to the actual label distributions) on the Cora dataset of non-overlapping and overlapping node scenarios with 20 clients, which are reported in Table 8. Specifically, as shown in Table 8, compared to the One scheme that uses only one node for calculating the functional embeddings, SBM and ER schemes that use more large numbers of randomly initialized nodes can accurately capture the similarities between subgraphs. This result demonstrates that a sufficient amount of randomness is required to capture the model's functional space. Also, compared to the Feature scheme that uses existing node representations to calculate the functional embeddings, SBM and ER random models show superiority in capturing similarities among subgraphs, which verifies that randomness indeed helps obtain accurate functional embeddings of models without incurring bias.

## C.7    RESULTS ON VARYING THE SIMILARITY ESTIMATION SCHEMES

As shown in Figure 3, our functional embeddings are not only effective but also efficient in capturing similarities between subgraphs, compared against using the parameter and gradient similarities. Additionally, while one might consider using the label distributions as the proxy for similarity estimation between clients, since labels are private local data stored in the client, this scheme may violate the privacy constraint of FL. However, to see their actual performances in the real-world dataset, we additionally conduct experiments on the parameter, gradient, and label similarities, on the Cora dataset of the overlapping node scenario with the number of clients

Table 9: Results on varying the similarity calculation schemes: parameter, gradient, label, and our functional embedding, on the overlapping node scenario with 30 clients of the Cora dataset.

| Model | Rounds | | | |
|---|---|---|---|---|
| | 20 | 40 | 60 | 80 |
| FedAvg | 29.94 | 32.69 | 47.84 | 52.42 |
| Parameter | 29.94 | 35.89 | 47.03 | 52.28 |
| Gradient | 33.93 | 51.09 | 52.77 | 58.14 |
| Label | 65.97 | 74.31 | 76.50 | 76.82 |
| Function (FED-PUB) | 67.82 | 73.51 | 74.66 | 75.90 |

as 30, and then compare the results with our functional similarities at 20, 40, 60, and 80 rounds.

As reported in Table 9, we can observe that the models, which utilize the parameter and gradient for calculating the similarities between subgraphs, show comparable performance to the naive FedAvg model, and inferior than our functional and label similarity schemes. However, even though the label similarity model uses privacy-sensitive local information (i.e., label distributions of every client), the performance of our FED-PUB that utilizes the functional embeddings from the privacy-free random

Table 10: Results on the overlapping node scenario with 10 clients (top) and non-overlapping node scenario with 30 clients (bottom), where we report results with mean and standard deviation over three different runs.

| Methods | Cora | CiteSeer | PubMed | Computer | Photo | obgn-arxiv |
|---|---|---|---|---|---|---|
| *Overlapping Node Scenario* | | | | | | |
| FED-PUB with Explicit Community | 80.45 ± 0.73 | 69.50 ± 0.20 | 84.76 ± 0.14 | 90.31 ± 0.06 | 92.67 ± 0.08 | 64.56 ± 0.12 |
| FED-PUB with Implicit Community | **81.54** ± 0.12 | **72.35** ± 0.53 | **86.28** ± 0.18 | **90.55** ± 0.13 | **92.73** ± 0.18 | **66.58** ± 0.08 |
| *Non-Overlapping Node Scenario* | | | | | | |
| FED-PUB with Explicit Community | **76.59** ± 0.39 | 67.57 ± 0.51 | 83.20 ± 0.45 | 87.84 ± 0.42 | 91.26 ± 0.25 | 61.52 ± 0.06 |
| FED-PUB with Implicit Community | 75.40 ± 0.54 | **68.33** ± 0.45 | **85.16** ± 0.10 | **89.15** ± 0.06 | **92.01** ± 0.07 | **63.34** ± 0.12 |

graph is similar to the performance of the label model. Therefore, along with the results in Figure 6, these comparison results on similarity schemes further verify the effectiveness of our functional embedding scheme in capturing the similarities among subgraphs, for identifying their communities.

## C.8    ANALYSES ON IMPLICIT AND EXPLICIT COMMUNITIES FOR WEIGHT AGGREGATION

As formalized in Equation 4 and described in Section 4.1, for personalized weight aggregation based on the functional similarities between clients, we implicitly model the community structures by performing weight aggregation over all available clients. However, one can alternatively perform explicit weight aggregation, by grouping similar subgraphs within the community first and then performing weight aggregation among clients within the community. To see which strategy is superior, we compare the performances of our variants: implicit and explicit community detection for weight aggregation. Note that, for the implicit setup, we use the formulation defined in Equation 4 without any modification. Meanwhile, for the explicit setup, we exclusively perform weight aggregation between clients, having the functional similarity score above 0.5, which we regard as forming the community, with the same normalization trick in Equation 4 after identifying communities.

As shown in Table 10, we observe that the model, which implicitly captures the community structures during weight aggregation, consistently outperforms the other explicit one, except for only one case: Cora with the overlapping node scenario. We believe such the exceptional case on the Cora dataset with the overlapping node scenario might be because, the information in the other communities is especially not useful for this particular setup; therefore, completely ignoring them contributes to the improved performance. Except for this, the results in Table 10 confirm that implicit modeling of community structures is generally better for personalized weight aggregation in subgraph FL.

## C.9    IMPACTS OF MISSING EDGES TO THE PERFORMANCE DEGENERATION

In this subsection, we empirically demonstrate that, due to the missing edge problem, all FL methods, which observe edges only within each subgraph, show inferior performances than the Oracle method, which trains on the entire graph including missing edges. To validate this claim, we first train the Oracle model on the connected global graph, and then evaluate it on disjoint subgraphs over all clients, on the Cora dataset of both Non-overlapping and Overlapping node scenarios with varying the client numbers. As

Table 11: Results on Non-Overlapping and Overlapping node scenarios with varying the number of clients on Cora. The Oracle model is not comparable, which trains with the global graph including missing edges.

| Model | Non-overlapping | | Overlapping | |
|---|---|---|---|---|
| | 5 Clients | 20 Clients | 10 Clients | 50 Clients |
| Oracle | 85.07 | 85.47 | 85.08 | 85.28 |
| Local | 81.30 | 80.30 | 73.98 | 76.63 |
| FedAvg | 74.45 | 69.50 | 76.48 | 53.99 |
| FedGNN | 81.51 | 70.10 | 70.63 | 56.91 |
| FedSage+ | 72.97 | 57.97 | 77.52 | 55.48 |
| FED-PUB (Ours) | 83.70 | 81.75 | 79.60 | 77.84 |

shown in Table 11, the Oracle model outperforms all the other methods, while our FED-PUB achieves the closest performance to the Oracle. The above results bring us to conclude that, due to the problem of missing edges, all FL methods, which trains with edges only within each subgraph, are inferior than the Oracle method. Note that this conclusion further suggests that the missing edge problem negatively affects the incompatible knowledge issue. Specifically, since all client models are trained on the partial subgraphs, which are parts of the larger global graph, the trained parameters in the client and the aggregated parameters in the server might not capture globally meaningful knowledge or the knowledge that is helpful to the other clients, while the Oracle model can capture.

## D  DISCUSSION ON LIMITATIONS AND POTENTIAL SOCIETAL IMPACTS

In this section, we discuss the limitations and potential societal impacts of our work.

**Limitations**   While our personalized subgraph FL framework, namely FED-PUB, is generally applicable regardless of subgraph types (e.g., unipartite graphs or bipartite graphs), our experiments are mainly done with unipartite graphs, which are the most popular setups. However, the efficacy of our FED-PUB on the other types of graphs, such as bipartite graphs, would be interesting to investigate, which have but not been explored so far, and we leave this as future work.

**Potential Societal Impacts**   The FL mechanism is important for preserving user's privacy, and, while this mechanism is actively studied in image and language domains, it gets little attention in graphs. However, we believe that our work comprehensively investigates and sufficiently tackles unique challenges in subgraph FL, such as missing nodes, edges, and their community structures.

Then, the potential positive impact of our work on society is that, ours can contribute to various domains that utilize graph-structured data, such as social, recommendation, and patient networks. Note that we would like to emphasize the importance of our subgraph FL scheme, especially in social and recommendation networks. In the current real-world application, all user's interactions with other users in social networks and with other products in recommendation networks may be stored in the server. However, this may not preserve the user's privacy, but also has potential risks of user data leakage from the server, such that storing user's data in the server is not recommended by existing data protection regularizations such as GDPR [3]. Then, by applying our subgraph FL framework to this domain, we expect such problems could be alleviated by not storing user's interaction data in the server, but only sharing the locally trained machine learning models from client subgraphs.

However, the transmitted model parameters from the client to the server may hold privacy-sensitive information. While eliminating it is not the main focus of our work (i.e., we assume that model parameters are transmittable without compromising privacy as in many FL works (McMahan et al., 2017; Li et al., 2020; Arivazhagan et al., 2019)), the research community may need to put further effort on whether the model parameters are safe, and how to make them more safe if they are not.

---

[3]https://gdpr-info.eu/

