# OpenReview forum: "Personalized Subgraph Federated Learning"
_ICLR.cc/2023/Conference — Submitted to ICLR 2023_

### Official Review · Reviewer_jjc2 · 2022-10-21

**Confidence:** 4
**Correctness:** 3
**Technical Novelty And Significance:** 2
**Empirical Novelty And Significance:** 3
**Recommendation:** 5

**Clarity, Quality, Novelty And Reproducibility:**

Clarity: It is easy for me to follow the story. All the components of the proposed method is well defined and explained at least intuitively.

Quality: The proposed techniques are well-motivated. Their design seems reasonable, at least from an intuitive aspect. The research questions answered in experiments are tightly related to the proposed method. However, without a theoretical analysis, I cannot persuade myself that the subgraph measurement must be helpful for clustering the clients. BTW, the discussions of related works could be further supplemented to be more comprehensive.

Novelty: FED-PUB mainly consists of two main components: one is the subgraph measurement to promote cluster-based personalized federated learning; another is the adaptive masking. The measurement utilizes the similarity in functional space, which is novel to me, in the jargon of graph learning (actually, it has been extensively studied in general machine learning). However, the adaptive masking has been observed in recent personalized federated learning works.

Reproducibility: All the adopted datasets are publicly available. All the considered baselines have open-sourced implementation. The experimental protocols and results are detailed.

**Strength And Weaknesses:**

Strengths:
1. This paper is well-written, with each figure, table, and paragraph being very self-contained.
2. The proposed method, FED-PUB, is the first one to exploit the functional embeddings for measuring subgraph similarities. Meanwhile, the authors introduce the client-wise learnable mask to further personalized the GNN models, in addition to the subgraph similarity-based cluster-based pFL aggregation.
3. The empirical studies in this paper is solid, with SOTA federated graph learning algorithm as a baseline, public datasets, and in-depth analysis from various perspectives.

Weaknesses:
1. It seems that there is no theoretical analysis about the effect of similarity measure, not to mention the eventual generalization risk. Actually, the relationship between graph data and the model parameters is intricate. Although the similarity in functional space seems to be a reasonable proxy, why it is useful needs to be analyzed and answered.
2. The masking scheme is not that novel to personalized federated learning. Thus, the technical contribution(s) of this paper seems to be insufficient (only the subgraph similarity measurement).

**Summary Of The Paper:**

This paper studies the non-iid subgraphs of the clients in an FL course and proposes a novel pFL algorithm to tackle this challenge. The idea of measuring subgraph similarity via functional embeddings' similarities is novel, and the adaptive masking is synergistic and seems to be helpful. The authors have conducted extensive empirical studies, which confirms the effectiveness of FED-PUB and the rationale behind its components.

**Summary Of The Review:**

This paper is completed and "standard". Techniques introduced in this paper are naturally motivated and sound helpful. However, the lack of a rigorous theoretical analysis makes me feel it is still below the bar of ICLR. I strongly encourage the authors to supplement such an indispensable part.

---

> ### Author Response · Authors · 2022-11-17
> **Initial response**
>
> We sincerely thank you for your constructive comments. We appreciate that you summarized our paper as completed and standard. Specifically, we thank you for your acknowledgments, in terms of 1) techniques that are well-motivated with reasonable design choices and tightly-related experiments; 2) novelty on functional embedding for subgraph similarities; 3) experimental results with in-depth analysis which are solid; 4) clarity where the paper is well-written with very self-contained contents. We initially address all your concerns:
>
> ---
>
> **Question 1:** Although subgraph similarities in the functional space seem to be a reasonable proxy, there is no theoretical analysis, which makes me feel below the ICLR bar.
>
> We appreciate your comment, however, we respectfully disagree, and we do not believe that every paper requires rigorous theoretical analysis to go over the ICLR bar. We agree with you that it is also useful to theoretically show the effectiveness of functional embeddings; however, we believe the theory is not the only way to show the benefits of functional similarities, and we already provided useful analyses on the efficacy of them empirically in Figure 6 and Figure 7 of the main paper, and Appendix C.6 and Appendix C.7 as well. In other words, we believe that, in science, empirical justification is one way of proving the effectiveness of the proposed methods, and we are confident that we sufficiently do this.
>
> On the other hand, it is extremely difficult to theoretically analyze the usefulness of functional embeddings in capturing similarities between subgraphs. In particular, we cannot leverage the supervised training signal about which subgraphs are more related to some other subgraphs, and also there are tons of uncontrollable parameters that, however, decide the functional embeddings of models. We can simply say that, as long as the subgraphs used for training are similar, the training objectives of their models are likely to be similar, and their trained models are similar in the functional spaces as well; however, as we explained, we do not have much information on which subgraphs are similar with training objectives and how tons of parameters are optimized, which make rigorous theoretical analysis difficult.
>
> ---
>
> **Question 2:** While the subgraph similarity measured in the functional space is novel, the adaptive masking, used in recent personalized FL, is not that novel; therefore, technical contributions only on the similarity measurement seem to be insufficient.
>
> Thank you for acknowledging the novelty of the functional similarities. On the other hand, for the novelty of the adaptive weight masking, even though it may be reminiscent of the other works in different FL domains (e.g. images), we believe such a point does not compromise our main novelty. Please note that our main novelty comes primarily from proposing and tackling the novel research problem on subgraph FL. Specifically, and, to the best of our knowledge, we firstly propose an undiscovered challenge of community structures, along with a missing edge problem, for subgraph FL tasks, illustrated in Figure 1 and formalized in Equation (2), which is significantly important as it largely degenerates the performance of existing subgraph FL methods. And, to tackle such unique challenges on subgraph FL, we propose to use subgraph similarity matching and personalized weight masking schemes; therefore, they are ingredients for dealing with the novel proposed problem. In other words, we believe our work is sufficiently novel, since the proposed problem of personalized subgraph FL is novel, and we propose effective and efficient solutions, including the subgraph measurement which is also novel, to tackle our novel problem.
>
> ---
>
> **Question 3:** The discussions of related works could be further comprehensive.
>
> Due to the page limits, we described details of the most related works to ours, which we also used for comparisons, comprehensively in Appendix B.2. We will provide more details in the Related Work section if an additional page is allowed after the acceptance.

---

> > ### Comment · Reviewer_jjc2 · 2022-11-26
> > **Discussion**
> >
> > Thanks for your response!
> >
> > Q1:
> > > We can simply say that, as long as the subgraphs used for training are similar, the training objectives of their models are likely to be similar, and their trained models are similar in the functional spaces as well;
> >
> > Yes, no theoretical analysis is ok to me. Besides, I also hold the same intuition with you, that is why I give "marginally below the acceptance threshold" rather than a rejection like the above reviewer. BTW, I just reviewed several papers on federated learning for recommendation, which happen to show counter examples regarding this intuition. This is just a "FYI", which I would not take into account due to they are not regular publication now.
> >
> > Q2:
> > > we firstly propose an undiscovered challenge of community structures, along with a missing edge problem, for subgraph FL tasks, illustrated in Figure 1 and formalized in Equation (2), which is significantly important as it largely degenerates the performance of existing subgraph FL methods.
> >
> > To be honest, I have not fully pinpointed what the proposed challenges exactly are. To my knowledge, FedSage+ [NeurIPS21] introduce the missing inter-client edges; FS-G [KDD22] discuss the non-iidness from both attributive and structural aspects, where client-wise graphs could have different homophilic levels yet the same node feature distributions. These are uniqueness of fed graph learning, which distinguishes fed graph learning from fed learning on regular euclidean data.

---

> > > ### Author Response · Authors · 2022-11-27
> > > **Further response**
> > >
> > > Thank you for your follow-up response, and we really appreciate it. The remaining concern that you have is, the uniqueness of this work on subgraph FL is unclear, and we further faithfully address it as follows:
> > >
> > > ---
> > >
> > > **Questions:** I have not fully pinpointed what the proposed challenges exactly are. FedSage+ [NeurIPS21] and FS-G [KDD22] have uniqueness, which distinguishes graph FL from regular FL on euclidean data. What is the uniqueness of this work?
> > >
> > > As clearly described and formalized in Section 3. Problem Statement, we focus on challenges in subgraph FL: network homophily and community structure in graph-structured data, and formalize them with personalized subgraph FL in Equation (2), which is different from what regular FL work considers.
> > >
> > > In particular, as described in the "Challenges in Subgraph FL" paragraph of Section 3, due to the network homophily, which is unique for graph-structured data, connected subgraphs have similar properties while others are not, and such property degenerates the performance of existing subgraph FL due to incompatible knowledge collapse. Therefore, from this motivation and observation, we formalize the subgraph FL problem with its inherent community structures arising from network homophily in graph-structured data, which is explained in the "Personalized Subgraph FL" paragraph of Section 3, and we show such the formalization prevents the knowledge collapse issue in subgraph FL, unlike existing subgraph FL.
> > >
> > > ---
> > >
> > > We hope the above response clarifies your concern about the uniqueness of our work in terms of graph FL, different from regular FL on euclidean data. Please let us know, if you have anything else that we should address further.

---

> > > > ### Comment · Reviewer_jjc2 · 2022-12-12
> > > > **About your question**
> > > >
> > > > I am glad to see that your paper studies subgraph FL, which introduces uniqueness against regular data such as images, sentences, and molecular graphs in the sense that whether training instances are independent.
> > > >
> > > > To my knowledge, FedSage+ is the first paper to complete the missing edges between subgraphs. As for FS-G, they introduce a synthetic FedcSBM, which includes the scenario, where the node attributes could follow the same distribution while the homophilic levels are different among the clients. It seems that APPNP, GPRGNN, BernNet, ChebNet, JacobiNet, etc. (decouple feature transformation from propagation) are good at handling such heterogeneity.
> > > >
> > > > I have not validated that point by myself, but it is sound from the formulation of those GNN.

---

### Official Review · Reviewer_PHC3 · 2022-10-21

**Confidence:** 4
**Correctness:** 3
**Technical Novelty And Significance:** 2
**Empirical Novelty And Significance:** 2
**Recommendation:** 3

**Clarity, Quality, Novelty And Reproducibility:**

a)	Clarity: this paper is well-written and easy to follow.
b)	Quality: the quality of this paper is good.
c)	Novelty: the idea is quite incremental, which limits the novelty of this paper.
d)	Reproducibility: The authors have attached the source codes in the supplementary file,


**Strength And Weaknesses:**

Pros:
	The motivation of this paper, i.e., considering the heterogeneity among subgraphs in the subgraph federated learning, is reasonable and has rarely been explored.
	The ideas are technically sound.
	Extensive experiments show the superiority of the proposed method compared to other subgraph federated learning methods.

Cons:
a)  The idea that performs subgraph federated learning among similar subgraphs is relatively incremental, limiting this paper's novelty.
b)  Some notations are confusing such as θ vs θ_i on page 4.
c)  The functional embeddings are key ingredients in the proposed FED-PUB, and the authors utilize a randomly sampled graph as the input of GNNs at each client. However, the influence of different random graphs is missing.
d)  At the beginning of model training, the functional embeddings derived from each client may not be accurate, resulting in the inaccurate grouping of communities. How does the proposed method deal with this situation?
e)  The description of baselines is incorrect. For example, FedPerGNN/FedGNN actually introduces a trusted third-party server to augment subgraphs, rather than based on the similarity between the nodes in the local client and the nodes in other clients. Furthermore, FedGNN is a method based on heterogeneous user-item bipartite graphs, and it is unclear how the authors apply it to homogeneous graphs (e.g., Cora). Therefore, it is better to provide a detailed description of experimental settings.


**Summary Of The Paper:**

Unlike the existing subgraph federated learning methods that overlook the heterogeneity among subgraphs, this paper proposes a personalized subgraph federated learning framework named FED-PUB, which enhances collaborative learning of subgraphs within the same community. To this end, it captures the communities based on functional embeddings of clients, then employs personalized sparse masks on the local GNN. Empirical results on six public datasets demonstrate the effectiveness of the proposed method.

**Summary Of The Review:**

Due to the limited technical novelty and misleading descriptions of related works, I recommend reject.

---

> ### Author Response · Authors · 2022-11-17
> **Initial response (2/2)**
>
> **Question 4:** At the beginning of model training, the functional embeddings may result in the inaccurate grouping of communities. How do the authors deal with this situation?
>
> We already showed and discussed the functional embeddings obtained in the first few rounds in the Community Detection paragraph of Section 5.2, and they can rapidly capture community structures in the first few rounds. Specifically, as shown in Figure 6, for the first few rounds, the model captures obvious communities first, i.e., the most similar subgraphs within the obvious community are grouped together at the beginning of FL. Then, at the later stages (e.g., 30 rounds), the model further captures less obvious communities, consisting of high as well as less similar subgraphs simultaneously, and the detected communities become very similar to the target similarities measured based on missing edges and label distributions. The reason the functional similarities can capture the community even at the beginning of FL might be because, the local models trained by similar subgraphs can probably have similar optimization objectives in every FL round including the first few; therefore, such the similar optimization objectives make the high functional similarities between models of similar subgraphs within the community, even in the beginning of training.
>
> ---
>
> **Question 5-1:**  The description of FedGNN (FedPerGNN) [1, 2] is incorrect. They introduce a trusted third-party server to augment subgraphs, instead of using similarities between nodes in different clients.
>
> The description of FedGNN (FedPerGNN) is correct, and they do use similarities between the nodes in the local client and the nodes in the other clients, before augmenting subgraphs. Let us more specifically describe that this work augments the particular client’s subgraph with nodes in the other clients, in which they propose to augment only the related nodes based on the similarity. And, the similarity is defined by overlapping nodes, e.g., if two nodes over two individual clients have shared neighboring nodes, they are transmitted and augmented.
>
> As you pointed out, they also additionally use the trusted third-party server for node sharing; however, as discussed in Section 3.5 of FedGNN [1], this node-sharing approach still has the risk of information leakage, but also has the inefficiency issue in communication costs shown in Table 3 of our paper.
>
> ---
>
> **Question 5-2:** FedGNN (FedPerGNN) is a method based on user-item bipartite graphs, and it is unclear how the authors apply it to homogeneous graphs.
>
> FedGNN (FedPerGNN) is not restricted to user-item bipartite graphs, since the core algorithms of it are to augment the nodes in the other subgraphs with privacy-preserving manners, and then perform the FedAvg to share model gradients/weights between clients. Therefore, to adapt this model to our experimental settings, we use the same GNNs as all compared models and use the proper node embedding/classifier layers for classification tasks, while measuring the similarities between nodes for augmentation with node features. We have clarified the experimental setup for FedGNN in Appendix B.2 during the revision.
>
> ---
>
>  ### References
>
> [1] Wu et al., Fedgnn: Federated graph neural network for privacy-preserving recommendation, KDD 2021.
>
> [2] Wu et al., A federated graph neural network framework for privacy-preserving personalization, Natura Communications 2022.

---

> ### Author Response · Authors · 2022-11-17
> **Initial response (1/2)**
>
> We sincerely thank you for your constructive comments. We appreciate that you acknowledged the strengths of our work, in terms of 1) motivation: the considered heterogeneity problem is reasonable and has rarely been explored; 2) ideas: the proposed approaches are technically sound; 3) experiments: we show the superiority of our method; also, 4) clarity, quality, and reproducibility. We initially address all concerns below:
>
> ---
>
> **Question 1:** Subgraph FL among similar subgraphs is relatively incremental, limiting the novelty.
>
> Our work’s novelty is not solely limited to calculating similarities between subgraphs.
>
> First of all, please note that our main originality comes primarily from proposing and tackling the novel research problem on subgraph FL. Specifically, and, to the best of our knowledge, we propose a novel challenge of community structures, along with a missing edge problem, for subgraph FL tasks, illustrated in Figure 1 and formalized in Equation (2). This problem is significantly important since it largely degenerates the performance of existing subgraph FL methods, and we propose to tackle this new problem with subgraph similarity matching and personalized weight masking schemes. In other words, since we aim to tackle the novel problem with valid approaches, our work’s novelty is not limited by only similarity learning.
>
> Also, subgraph FL among similar subgraphs itself has a novel contribution. In particular, subgraph FL among similar subgraphs is the objective to tackle community-structural challenges in subgraph FL, which are rarely explored as you acknowledged; and, for this problem, the main contribution comes from how to measure subgraph similarities in FL settings, which is extremely challenging since we cannot know which subgraph each client has. To this end, we propose a novel method that calculates the similarities between clients with their models’ functional embeddings, and we believe this approach is also highly novel.
>
> ---
>
> **Question 2:** Some notations are confusing such as $θ$ vs $θ_i$ on page 4.
>
> Thanks for pointing it out. $\theta$ denotes the global parameters, and $\theta_i$ denotes the local parameters for the i-th client. To further clarify, we have revised the notation for the global parameters $\theta$ to $\bar \theta$.
>
> ---
>
> **Question 3:** To compute functional embeddings, the authors use a randomly sampled graph; however, the analysis of different random graphs for functional embeddings is missing.
>
> We already provided the influences of different random graphs for functional embeddings in Appendix C.6 and Appendix C.7. We also refer to them in Section 4.1: “We provide additional discussions with results on similarity estimation in Appendix C.6 and C.7”.

---

> ### Author Response · Authors · 2022-11-27
> **Further response**
>
> Thank you for your follow-up. We are more than happy that our initial response clear-ups your concerns and comments, and the remaining only concern is about the novelty. In this response, we address it, as below:
>
> ---
>
> **Question:** The research problem of learning subgraph similarities in FL settings is relatively incremental.
>
> Our work is clearly novel and not incremental, in terms of the following aspects:
> * Considering the community structure for subgraph FL is novel, which also leads to significant performance gains in subgraph FL.
> * Tacking the community structure problem with similarities between subgraphs is novel.
> * Proposing the functional embeddings for capturing community structures (i.e., subgraph similarities) is novel.
>
> To the best of our knowledge, the above aspects have not been explored so far. Could you let us know, if you know related works that handle the above, so that we can discuss and compare them?
>
> ---
>
> Also, please note that the novelty comes from "how to capture similarities", not merely from "using similarities". For instance, in general, there are tons of machine learning work that proposes to capture similarities, for example, attention mechanism, siamese neural network, or semi-parameter approaches, to name a few, and they are considered novel since they propose a new way of capturing similarities, as ours that proposes functional embeddings.

---

### Official Review · Reviewer_TrJG · 2022-10-27

**Confidence:** 3
**Correctness:** 2
**Technical Novelty And Significance:** 2
**Empirical Novelty And Significance:** 2
**Recommendation:** 3

**Clarity, Quality, Novelty And Reproducibility:**

A few comments on paper writing:
1. In Section 3 "each column represents a ..." -> "each row represents a ...".
2. Section 3 uses superscripts with brackets for both rounds and layers.
3. In Section 3, "few existing methods" -> "existing methods".
4. In Equation (2), what does it mean by summing over G_i when optimizing (\theta_i, \mu_i)?
5. Figure 2 caption: "each of which consists of one/two subgraphs" -> "community A consists of one subgraph, and community B consists of two subgraphs."

**Strength And Weaknesses:**

Strengths:
1. This paper proposes an implicit method to exploit the community structure within the graph using pairwise similarity scores on clients. With this method, we can properly handle missing edges and subgraph heterogeneity with a reasonable performance gain in accuracy.

2. Some implementation refinements, such as masking and regularizations, are applied to achieve better performance.

Weaknesses:
1. The method may sacrifice the user's privacy by calculating personalized information on the server side. From Equation (4), even though the information among different clients are mixed together on the server's side, there is a chance that personal data can leak since the mixing rule encourages aggregating similar clients. However, the paper needed to articulate the tradeoff between privacy and accuracy.

2. The paper should have explicitly modeled a rigid community structure on the server's side. Instead, it uses an implicit similarity measurement that can potentially measure more complicated structures, such as overlapping communities. However, the empirical analysis needed to show a deeper insight using the similarity structure.

**Summary Of The Paper:**

This paper proposes a personalized method for subgraph learning with multiple clients in different communities. The key idea is to measure the similarity between each customer pair and use that similar information for succeeding updates. Empirical results demonstrate superior performance on test accuracy.

**Summary Of The Review:**

This paper proposed a personalized idea for subgraph federated learning. My concern is that the accuracy gain may come from user information leakage. And there is no systematic discussion regarding the privacy-accuracy tradeoff.

---

> ### Author Response · Authors · 2022-11-17
> **Initial response (2/2)**
>
> **Question 3.1:** In Equation (2), what does it mean by summing over $G_i$ when optimizing ($\theta_i$, $\mu_i$)?
>
> Thank you for pointing it out. We have corrected the notation as follows: summing over $G_i$ when optimizing { $\theta_i, \mu_i$ } $_{i=1}^K $, where $K$ is the number of all clients participating in FL.
>
> ---
>
> **Question 3.2:** Few other comments on paper writing.
>
> Thank you for your corrections, and we have revised all of them during the revision.
>
> ---
>
>  ### References
>
> [1] Sattler et al., Clustered Federated Learning: Model-Agnostic Distributed Multi-Task Optimization under Privacy Constraints, IEEE Transactions on Neural Networks and Learning Systems 2020.
>
> [2] Beaussart et al., WAFFLE: Weighted Averaging for Personalized Federated Learning, NeurIPS 2021 Workshop.
>
> [3] Xie et al., Federated Graph Classification over Non-IID Graphs, NeurIPS 2021.
>
> [4] McMahan et al., Communication-Efficient Learning of Deep Networks from Decentralized Data, AISTATS 2017.

---

> ### Author Response · Authors · 2022-11-17
> **Initial response (1/2)**
>
> We sincerely thank you for your constructive comments. We appreciate that you acknowledged our contributions of 1) leveraging community structures to handle missing edge and subgraph heterogeneity problems; 2) proposing masking and regularization techniques for performance improvements. We initially address all of your concerns below:
>
> ---
>
> **Question 1-1:** Aggregating similar clients in the server may sacrifice the user’s privacy.
>
> Our similarity-based personalized aggregation in the server does not sacrifice the user’s privacy. Following many FL works [1, 2, 3], as long as we do not send the local data to the server, we assume the user's privacy is preserved, which is also described in the first FL paper (i.e., FedAvg) [4]. In our case, we can only share only the parameters with the server, and then obtain the functional embeddings in the server, which are then used for calculating the similarities between subgraphs. In this process, there is no usage of the local data, and, since we only use the model parameters, the privacy constraint is the same across all the other compared methods including ours which perform parameter aggregation in the server.
>
> ---
>
> **Question 1-2:** The accuracy gain may come from user information leakage.
>
> Since there is no user information leakage (See **Question 1-1**), the accuracy gain of our model does not come from user information leakage. The accuracy gain rather comes fairly from the effectiveness of our training objective formalized in Equation (2), and our model architectures: similarity matching and weight masking, described in Sections 4.1 and 4.2.
>
> ---
>
> **Question 1-3:** There is no systematic discussion on the privacy-accuracy tradeoff.
>
> Since we do not sacrifice the user’s privacy (See **Question 1-1**), the privacy-accuracy tradeoff does not occur in our framework, and the discussion on it is also not necessary.
>
> ---
>
> **Question 2:** This work implicitly models the community structure among subgraphs, instead of explicitly modeling it; however, there is no empirical analysis of this design choice.
>
> Thank you for pointing it out. As explained in the “Personalized Weight Aggregation Based on Subgraph Similarity” paragraph of Section 4.1, we implicitly model the community structure by performing weight averaging over all clients, and this is because, when we conducted initial experiments, we observed that explicitly modeling the community structure results in the suboptimal performance. During the response period, to concretely show the advantage of the implicit modeling of community structures, we have compared the performances of our FED-PUB between the explicit community and the implicit community setups. As shown in Table A and Table B below, we observe that the model, which implicitly captures the similarities for community detection for weight aggregation, consistently outperforms the other explicit one, except for only one case: Cora with the overlapping node scenario. This result confirms that implicit modeling of community structures is generally better for personalized weight aggregation in subgraph FL. We have included the discussion on it with more detailed experimental setups in Appendix C.8, during the revision.
>
>
> Models | Cora | Citeseer | Pubmed | Amazon-Computer | Amazon-Photo | obgn-arxiv
> ------------- | ------------- | ------------- | ------------- | ------------- | ------------- | -------------
> FED-PUB with Explicit Community  | 80.45 ± 0.73 | 69.50 ± 0.20 | 84.76 ± 0.14 | 90.31 ± 0.06 | 92.67 ± 0.08 | 64.56 ± 0.12
> FED-PUB with Implicit Community  | 81.54 ± 0.12 | 72.35 ± 0.53 | 86.28 ± 0.18 | 90.55 ± 0.13 | 92.73 ± 0.18 | 66.58 ± 0.08
>
> Table A: Results on the overlapping node scenario with 10 clients, where we report results with mean and standard deviation over three different runs.
>
>
> Models | Cora | Citeseer | Pubmed | Amazon-Computer | Amazon-Photo | obgn-arxiv
> ------------- | ------------- | ------------- | ------------- | ------------- | ------------- | -------------
> FED-PUB with Explicit Community  | 76.59 ± 0.39 | 67.57 ± 0.51 | 83.20 ± 0.45 | 87.84 ± 0.42 | 91.26 ± 0.25 | 61.52 ± 0.06
> FED-PUB with Implicit Community  | 75.40 ± 0.54 | 68.33 ± 0.45 | 85.16 ± 0.10 | 89.15 ± 0.06 | 92.01 ± 0.07 | 63.34 ± 0.12
>
> Table B: Results on the non-overlapping node scenario with 30 clients, where we report results with mean and standard deviation over three different runs.

---

### Author Response · Authors · 2022-11-26
**Reminder**

Dear reviewers,

We sincerely appreciate your time and effort in reviewing our paper. We initially address all your comments, and could you go over our response, so that we can address further, if there exist remaining concerns?

---

### Author Response · Authors · 2022-12-05
**Gentle Reminder**

Dear reviewers,

The end of the discussion period is fastly approaching, and could you please go over our response? We strongly believe our last response clear-ups your concerns/comments, and we look forward to receiving updates from you soon.

Many thanks again for reviewing our paper.

---

### Decision · Program_Chairs · 2023-01-20

**Decision:**

Reject

**Justification For Why Not Higher Score:**

There are several concerns preventing the acceptance, especially on whether the improvement is at the cost of privacy. The authors fail to clarify this in the response.

**Justification For Why Not Lower Score:**

N/A

**Metareview: Summary, Strengths And Weaknesses:**

This paper introduces a subgraph-similarity-based federated learning framework. Novelties include using subgraph similarities to guide the FL update, and using functional embeddings to measure subgraph similarities. I would like to authors know that I have carefully read all the reviews and author responses. Firstly, I think reviewer Reviewer TrJG's concern that using subgraph-similarity to compute personalized parameters may leak privacy is a valid point. This is because the information from different clients are unevenly mixed in the server and returned to individual client. In the extreme case one client might be able to fully recover the other client's parameters when this other client is dominantly similar. Secondly, I hold the same opinion as reviewer PHC3 that the novelty of modeling subgraph similarities to augment FL is somewhat incremental, and the way of using one random graph as input to two GNNs to measure their similarities is questionable. If we want to measure the similarities between two functions, we either compare their parameters (for parameterized functions) or evaluate their function values on a wide range of inputs (for nonparameterized functions). I doubt that evaluating only on a single random graph will be highly inaccurate. The authors are encouraged to discuss the above points in their revision.

---

> ### Author Response · Authors · 2023-02-06
> **Thank you for meta-reviewing our work, and we would like to leave comments on the meta-review**
>
> Dear Chairs,
>
> Thank you for your time and effort in meta-reviewing our paper, and, of course, we accept your final decision. However, we would like to politely clarify some of the misunderstandings.
>
> First, you mentioned that performing personalized weight aggregation with regard to functional similarities of GNN parameters may leak privacy since parameters are unevenly mixed in the server. However, we strongly believe that an uneven mixture of parameters makes the clients more difficult to recover other clients' parameters. This is because, to recover them, each client should further know the coefficients for weight aggregation, as well as the parameters in the other clients. However, they are not accessible in our FL scenario. Moreover, you provided the extreme case where the functional similarities of two clients are dominantly similar, and they substantially share model weights, in which you argued that one client may recover parameters from the other client. However, local clients cannot know the other clients that they share weights substantially, and also we do not perform such exclusive weight aggregation (Equation 4); therefore, local clients cannot exactly recover the parameters from the other clients. Finally, in the existing FL literature [1, 2, 3], clustering-based weight aggregation is widely used, and they share a similar spirit in weight averaging to ours. Therefore, we believe our weight averaging mechanism does not sacrifice the user's privacy.
>
> Second, you pointed out that the way of using one random graph as input to two GNNs to measure their similarities is questionable since similarities based only on a single random graph will be highly inaccurate. However, we indeed used five individual subgraphs, each of which consists of 100 nodes; thus, the number of features used for calculating functional similarities is 500 (See Appendix B.3). Also, we already provided analyses on random graph inputs for functional similarities in Appendix C.6, which also includes the similarity result with one node feature.
>
> We have further revised our manuscript, highlighting the suggested two points. Thank you.
>
> Best regards, Authors
>
> [1] An Efficient Framework for Clustered Federated Learning, NeurIPS 2020.
>
> [2] Clustered Federated Learning: Model-Agnostic Distributed Multi-Task Optimization under Privacy Constraints, IEEE Transactions on Neural Networks and Learning Systems 2021.
>
> [3] Federated graph classification over non-iid graphs, NeurIPS 2021.